# Follicular helper- and peripheral helper-like T cells drive autoimmune disease in human immune system mice

**Mohsen Khosravi-Maharlooei[1,2], Andrea Vecchione[1,3], Nichole Danzl[1], Hao Wei Li[1], Grace Nauman[1,4], Rachel Madley[1,4], Elizabeth Waffarn[1], Robert Winchester[1], Amanda Ruiz[1], Xiaolan Ding[1], Georgia Fousteri[3], Megan Sykes[1,4,5]\***

[1]Columbia Center for Translational Immunology, Department of Medicine, Columbia University Medical Center, Columbia University, New York, United States; [2]Department of Immunology, Department of Biochemistry and Molecular Biology, Mayo Clinic, Phoenix, United States; [3]San Raffaele Hospital, Milan, Italy; [4]Department of Microbiology and Immunology, Columbia University Medical Center, Columbia University, New York, United States; [5]Department of Surgery, Columbia University Medical Center, Columbia University, New York, United States

**\*For correspondence:**
ms3976@cumc.columbia.edu

## eLife Assessment

This **important** study utilizes humanized mice, in which human immune cells are introduced into immune-deficient mice, to provide **convincing** evidence that two helper CD4 T-cell subsets, T-follicular helper (Tfh) and T-peripheral helper (Tph) cells, are able to drive both autoantibody production and induction of autoimmunity. The work will be of broad interest to medical scientists engaged in deciphering how human immune cells mediate immune responses and contribute to the development of autoimmune diseases.

**Abstract** Human immune system (HIS) mice constructed in various ways are widely used for investigations of human immune responses to pathogens, transplants, and immunotherapies. In HIS mice that generate T cells de novo from hematopoietic progenitors, T cell-dependent multisystem autoimmune disease occurs, most rapidly when the human T cells develop in the native NOD.Cg-*Prkdc*scid *Il2rg*tm1Wjl (NSG) mouse thymus, where negative selection is abnormal. Disease develops very late when human T cells develop in human fetal thymus grafts, where robust negative selection is observed. We demonstrate here that PD-1+CD4+ peripheral (Tph) helper-like and follicular (Tfh) helper-like T cells developing in HIS mice can induce autoimmune disease. Tfh-like cells were more prominent in HIS mice with a mouse thymus, in which the highest levels of IgG were detected in plasma, compared to those with a human thymus. While circulating IgG and IgM antibodies were autoreactive to multiple mouse antigens, in vivo depletion of B cells and antibodies did not delay the development of autoimmune disease. Conversely, adoptive transfer of enriched Tfh- or Tph-like cells induced disease and autoimmunity-associated B cell phenotypes in recipient mice containing autologous human APCs without T cells. Tfh/Tph cells from mice with a human thymus expanded and induced disease more rapidly than those originating in a murine thymus, implicating HLA-restricted T cell-APC interactions in this process. Since Tfh, Tph, autoantibodies, and lymphopenia-induced proliferation (LIP) have all been implicated in various forms of human autoimmune disease, the observations here provide a platform for the further dissection of human autoimmune disease mechanisms and therapies.

## Introduction

Human immune system (HIS) mice provide unique opportunities to investigate human immune biology and therapies. Many different versions of these models exist, the simplest of which involves infusion of human lymphocytes from human donors into immunodeficient mouse recipients. Such models, however, are limited by graft-vs-host disease (GVHD) initiated by adoptively transferred human xenoreactive T cells that recognize murine host antigens, thereby limiting the duration and type of information that can be obtained from such models. Additional HIS mouse models involve the de novo generation in immunodeficient mouse recipients of HIS from human hematopoietic stem and progenitor cells injected intravenously. When NOD mice deficient in T, B, and NK cells such as NOD. Cg-*Prkdc*scid *Il2rg*tm1Wjl (NSG) mice are used as recipients, human T cells develop in the murine thymus, while B cells and myeloid antigen-presenting cells develop in the recipient bone marrow. However, we recently demonstrated that human T cells developing de novo in the NSG mouse thymus do not undergo normal negative selection for self-antigens and lack normal thymocyte diversity. We further showed that the thymus does not develop normal structure and contains a paucity of medullary epithelial cells. These T cells induce a multiorgan disease characterized by human immune cell infiltrates in multiple organs, usually within 5–8 months after transplantation (*Khosravi-Maharlooei et al., 2021*). Because the causal T cells develop de novo in the recipient mice, we describe this as an autoimmune disease rather than GVHD, which is induced by mature T cells transferred to immunodeficient mice in human hematopoietic cell grafts (*Brehm et al., 2019*). Following transplantation of an autologous or partially HLA-matched (to intravenously administered fetal CD34+ cells) human fetal thymus graft, the developing thymocytes undergo negative selection for self-antigens (*Khosravi-Maharlooei et al., 2021*), reflecting the presence of both human and murine (*Kalscheuer et al., 2012*) APCs in the thymus graft and the presence of a normal human cortico-medullary thymic structure (*Khosravi-Maharlooei et al., 2021*). T cells developing in a human thymus graft eventually cause autoimmune disease, but with a significant delay if the native murine thymus has been removed (*Khosravi-Maharlooei et al., 2021*). The slowly evolving autoimmune disease induced by T cells developing de novo in human thymus grafts is independent of direct recognition of murine antigens, as it develops with similar velocity in NSG mice expressing murine MHC and those completely lacking murine Class I and Class II MHC antigens (*Khosravi-Maharlooei et al., 2021*).

The original descriptions of this model involved the use of human fetal thymus and liver tissue engrafted under the kidney capsule along with intravenous administration of CD34+ cells from the same FL (*Lan et al., 2004*; *Lan et al., 2006*; *Melkus et al., 2006*). While the term 'bone marrow, liver, thymus, BLT' mouse has been widely used to describe this model, the model does not include bone marrow and we found the FL fragment to be unnecessary for immune system development. Consequently, 'BLT' does not adequately describe the essential components of the model. Instead, we have adopted the term 'Hu/Hu' to denote HIS mice constructed with human (Hu) fetal thymus and autologous human FL CD34+ cells administered i.v. Denoting the origin of the thymus and CD34+ cells separately has allowed us to specify variations on the model in which, for example, the thymus tissue originates in fetal swine (the 'Sw/Hu' model [*Nikolic et al., 1999*; *Shimizu et al., 2008*; *Khosravi Maharlooei, et al., 2017*; *Kalscheuer et al., 2014*]) or when the native mouse thymus is used to generate human T cells (the 'Mu/Hu' model described here) (*Khosravi-Maharlooei et al., 2024*).

We have now examined the role of B cells, autoantibodies, and T cell help for B cells in the development of autoimmune disease in HIS mice. Our results demonstrate that B cell help from follicular (Tfh)- and peripheral (Tph) helper-like T cells drives B cell differentiation and autoantibody responses and that these T cells are sufficient to cause disease in the absence of B cells or antibodies. Furthermore, a global lack of T cell tolerance to murine and human self-antigens is a major driver of autoimmune disease among HIS mice whose T cells develop in a murine thymus. In contrast, the disease that develops later in HIS mice whose T cells develop in a human thymus is likely dependent on a lack of tolerance to tissue-restricted murine antigens presented on human APCs. These studies provide novel insights with utility for dissecting mechanisms of autoimmune disease induction by human T cells.

## Results

### Mouse-reactive T cells in HIS mice with mouse or human thymus

We have previously demonstrated that a thymus-dependent multiorgan autoimmune disease occurs in HIS mice generated by intravenous injection of human fetal liver (FL) CD34$^+$ cells into NSG mice and that this disease develops more rapidly in mice containing a native murine thymus (termed 'Mu/Hu mice' [murine thymus/human CD34$^+$ cells]) than in thymectomized (*Khosravi-Maharlooei et al., 2020*) NSG mice receiving a human thymus graft, termed 'Hu/Hu mice' (human thymus/human CD34$^+$ cells) (*Khosravi-Maharlooei et al., 2021*). While Mu/Hu mice rely on the native mouse thymus for human T cell development, T cells in Hu/Hu mice develop in the human fetal thymus graft. At 20 weeks after transplantation, animals in both groups were sacrificed and their splenocytes were CFSE-labeled and

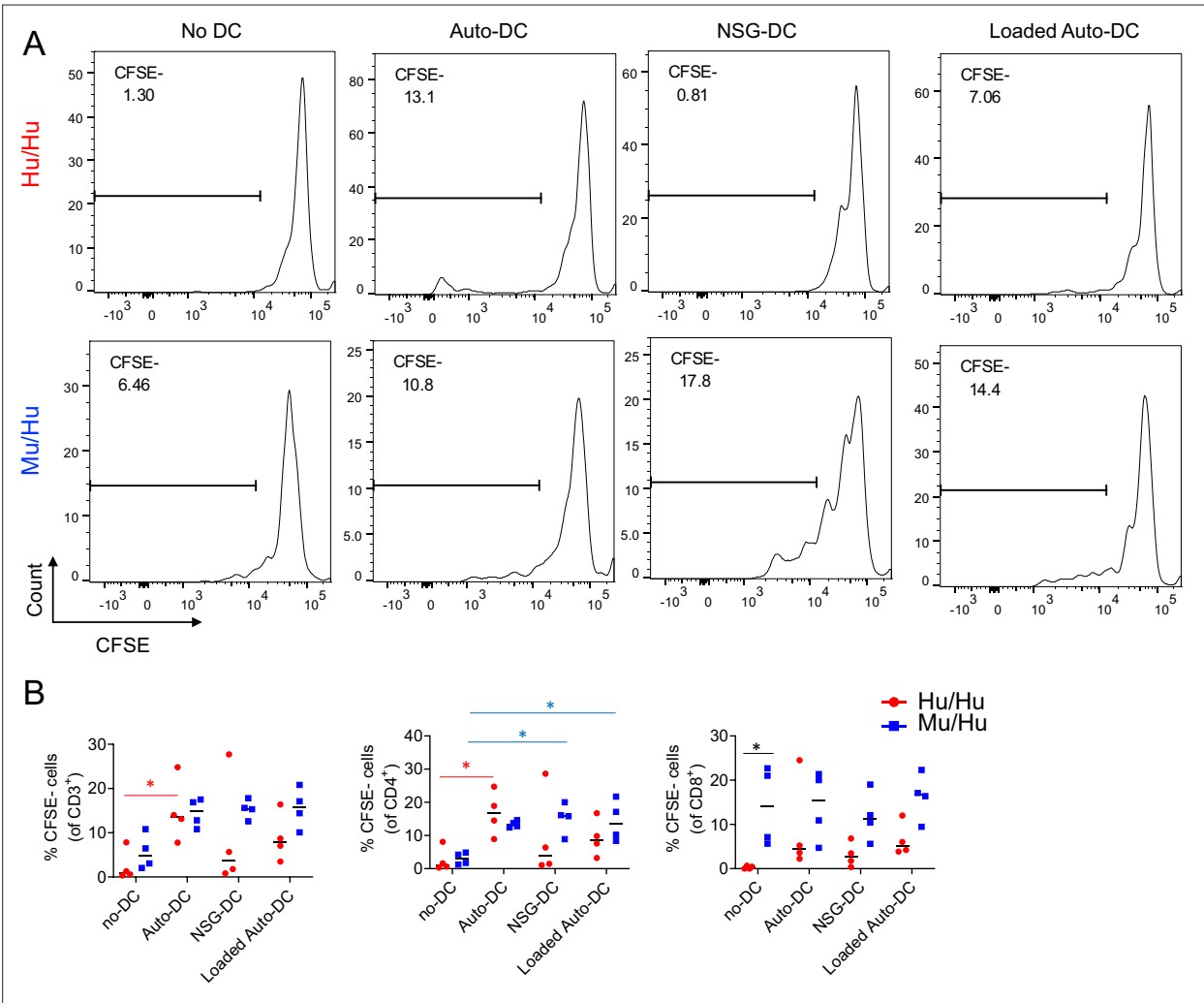

**Figure 1.** Lack of tolerance to murine recipient antigens of CD4 T cells developing in mouse thymus compared to those developing in human thymus. Mu/Hu (n=4) and Hu/Hu (n=4) mice were sacrificed 20 weeks after transplantation and their splenocytes were CFSE-labeled and tested for reactivity to various antigen-presenting cells. To test direct reactivity to autologous human dendritic cells (DCs), fetal liver (FL) CD34$^+$ cells used to generate both Hu/Hu and Mu/Hu mice were differentiated into DCs. NSG DCs were generated from bone marrow progenitors. Proliferation of T cells was measured after 6 days of co-culture based on CFSE dilution. (**A**) Representative plots showing proliferation of HuHu (top) and MuHu (bottom) T cells following co-culture with autologous human DCs, NSG mouse DCs, autologous human DCs loaded with murine antigens or with no DC. (**B**) Frequencies of proliferating CD3$^+$ T cells and CD4$^+$ and CD8$^+$ T cells from splenocytes of HuHu (red) and MuHu (blue) mice in response to the indicated DCs. Differences between proliferation rate of Hu/Hu and Mu/Hu T cells were analyzed with unpaired t-test. In all graphs, each point represents an individual mouse with the mean indicated by a black line. Asterisks indicate statistical significance. **p<0.01, *p<0.05 . Statistically significant differences in responses between Hu/Hu or Mu/Hu T cells compared to the same group of T cells stimulated with other DCs are indicated by red and blue asterisks, respectively, while differences between Hu/Hu and Mu/Hu T cells are marked with black asterisks.

tested for reactivity to autologous FL-derived human dendritic cells (DCs) and for responses to NSG mouse bone marrow-derived DCs (anti-host response) as well as responses to autologous human DCs pulsed with apoptotic NSG mouse DCs (indirect anti-host response). As shown in *Figure 1A and B*, T cells from Hu/Hu and Mu/Hu mice showed proliferation to autologous human DCs that was not significantly augmented by pulsing of human DCs with murine antigens. Since the responder splenocyte preparations were not depleted of murine cells, we cannot distinguish whether the baseline proliferation represents responses to human antigens or to murine antigens presented by these DCs. However, proliferation of both CD4 and CD8 T cells to NSG mouse DCs was greater in Mu/Hu mice than in three of four Hu/Hu mice, which showed little, if any, direct proliferation to murine antigens. Proliferation of Mu/Hu CD4$^+$ T cells to murine DCs was significantly greater than proliferation in the absence of DCs, in contrast to Hu/Hu CD4 cells, suggesting a lack of tolerance to NSG antigens only in Mu/Hu CD4$^+$ T cells (*Figure 1B*). This observation is consistent with the lack of cortico-medullary structure or normal negative selection in the native murine thymi (*Khosravi-Maharlooei et al., 2021*).

## Increased number of Tfh cells in HIS mice with a mouse thymus compared to those with a human thymus

Since increases in Tfh and Tph have been associated with various human autoimmune diseases, we compared Tfh-like and Tph-like cells in groups of Mu/Hu (n=8) and Hu/Hu mice (n=11) generated from the same two FL donors (*Figure 2A*). CD3$^+$ T cells were detected in peripheral blood around 12 weeks after transplantation (data not shown), as we previously reported (*Kalscheuer et al., 2012*; *Khosravi-Maharlooei et al., 2021*; *Khosravi-Maharlooei et al., 2020*).

We monitored PD-1$^+$CXCR5$^-$ Tph-like and PD-1$^+$CXCR5$^+$ Tfh-like CD4$^+$CD45RA$^-$ T cell reconstitution from 12 to 32 weeks post-transplantation (*Figure 2B–D*). Circulating Tph-like cells were detected at 13 weeks post-transplantation in both groups. The frequency and absolute numbers of Tph-like cells in blood of Mu/Hu mice tended to be greater than those in Hu/Hu mice and the difference in percentages achieved statistical significance across multiple time points (*Figure 2C*). Circulating Tfh-like cells appeared at 13 weeks post-transplantation in Mu/Hu mice (*Figure 2D*), increased progressively over time and were significantly more abundant in this group than in Hu/Hu mice (*Figure 2D*).

In additional experiments, we compared splenic Tfh-like and Tph-like cells in Mu/Hu (n=19) and Hu/Hu mice (n=29) (n=5 different FL donors) sacrificed before 20 weeks, between 20 and 30 weeks, and more than 30 weeks post-CD34$^+$ FL transplantation (*Figure 2E*). While no significant differences in Tph-like cell frequency and absolute numbers were detected between the two groups (*Figure 2F*), percentages of Tfh-like cells were significantly greater in spleens of Mu/Hu compared to Hu/Hu mice (*Figure 2G*). The numbers of both Tph-like and Tfh-like cells increased over time in both HIS mouse groups and staining for Ki67 revealed high proliferative rates for both cell types (*Figure 2—figure supplement 1*).

These results suggest that Tph and Tfh differentiate and expand over time and do so more rapidly in Mu/Hu than Hu/Hu mice. These data correlate with the more rapid onset of autoimmune disease in Mu/Hu than Hu/Hu mice, consistent with a role for these T cell types in disease development.

To further characterize Tph-like and Tfh-like cells in HIS mice, we measured ICOS expression and IL-21 production (*Figure 2H and I*). These cells expressed higher levels of ICOS (*Figure 2H*) and, upon phorbol myristic acid (PMA)/ionomycin stimulation, secreted more IL-21 (*Figure 2I*) compared to control, CD45RA$^+$ T cells, consistent with helper function and with their role in humans. The majority of Tfh-like and Tph-like cells in Mu/Hu and Hu/Hu mice expressed ICOS, though the percentage of ICOS$^+$ cells was significantly higher in Mu/Hu Tfh-like and Tph-like cells compared to their Hu/Hu counterparts (*Figure 2J*).

We also compared the levels of FOXP3 within splenic Tfh-like and Tph-like populations in Hu/Hu and Mu/Hu mice. Approximately 10% of Tfh-like and Tph-like cells in both Hu/Hu and Mu/Hu groups were FOXP3$^+$, with the highest proportion observed in Hu/Hu Tph-like cells (*Figure 2K*).

## Increased serum IgG in Mu/Hu compared to Hu/Hu HIS mice

To determine whether or not the increased proportion of Tfh-like cells in Mu/Hu HIS mice impacted B cell differentiation and survival, as reported in humans (*Weinstein et al., 2012*), we monitored total B cell reconstitution and serum IgM and IgG levels over time. While overall B cell reconstitution (*Figure 3A*) and serum IgM concentration was similar between Mu/Hu and Hu/Hu mice, serum IgG

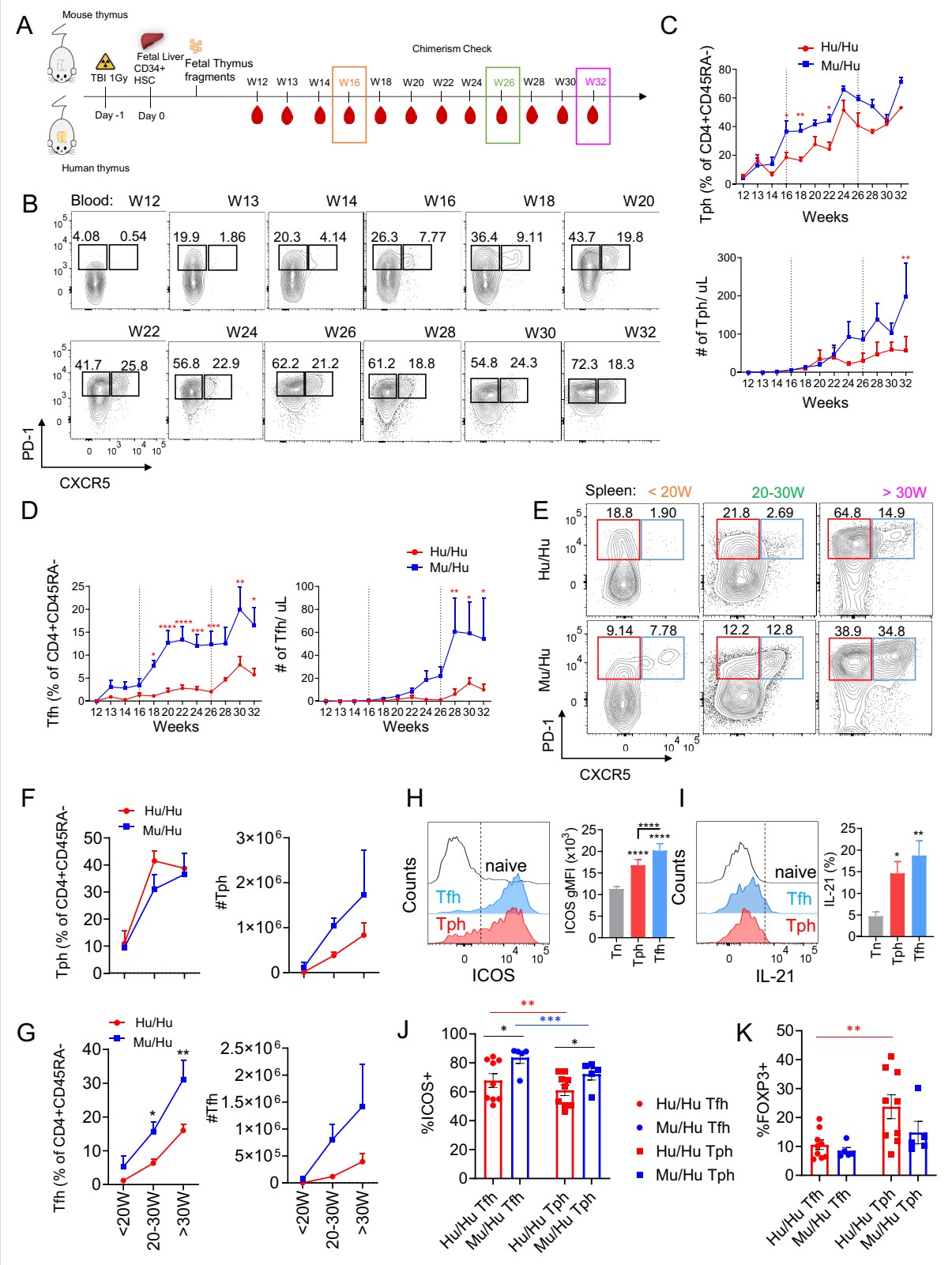

**Figure 2.** Tfh and Tph cell reconstitution in human immune system (HIS) mice. (**A**) Schematic of experimental design; (**B**) representative staining; (**C,D**) percentage of CXCR5-PD-1+ Tph and CXCR5+PD-1+ Tfh cells among CD4+CD45RA- T cells and absolute counts per microliter in peripheral blood of HIS mice with mouse thymus (n=19) or human thymus (n=29) from week 12 to week 32 post-transplantation, analyzed every 2 weeks; (**E**) representative staining and (**F,G**) percentage of CXCR5-PD-1+ Tph and CXCR5+PD-1+ Tfh cells among CD4 T cells and absolute counts in spleens of

*Figure 2 continued on next page*

Figure 2 continued

HIS mice with mouse thymus or human thymus at <20W, 20–30W, and >30W post-transplantation. (**H**) Expression of ICOS in Tph and Tfh from Mu/Hu (n=16) and Hu/Hu (n=8) mice (combined results) and (**I**) IL-21 cytokine production after PMA/ionomycin stimulation of Tph (CXCR5⁻PD-1⁺) and Tfh cells (CXCR5⁺PD-1⁺) compared to naïve CD4⁺ T cells (gray) from Mu/Hu (n=11) and Hu/Hu (n=4) mice (combined results). The results are expressed as mean ± SEM geometric mean fluorescence intensity (MFI) values; (**J,K**) percentage of ICOS⁺ and FOXP3⁺ cells among splenic Hu/Hu and Mu/Hu Tfh and Tph cells. All data are shown as means ± SEM. Asterisks indicate statistical significance between Hu/Hu and Mu/Hu groups as calculated by Bonferroni multiple comparison test. *p<0.05,. **p<0.01, ***p<0.001, and ****p<0.0001.

The online version of this article includes the following figure supplement(s) for figure 2:

**Figure supplement 1.** Ki67 expression in splenic Tph and Tfh cells at 30 weeks post-transplantation.

levels were significantly greater in Mu/Hu mice (*Figure 3B*), suggesting increased B cell differentiation and IgG class switching.

## Helper function of Tfh and Tph in Mu/Hu and Hu/Hu mice

Next, to compare the functional capacity of Tfh-like and Tph-like cells from HIS mice with mouse or human thymus, we measured the ability of sorted CD25⁻CXCR5⁺ CD45RA⁻ (including Tfh and excluding regulatory follicular cells) and CD25⁻ CXCR5⁻CD45RA⁻ (including Tph cells and excluding Tregs) CD4⁺ cells to induce differentiation of CD19⁺CD38⁻IgD⁺CD27⁻ naïve B cells to CD20⁻CD38⁺ plasmablasts in vitro (*Figure 3C*). After isolation, these T cell fractions were incubated with autologous naïve B cells. Helper cells from both Mu/Hu and Hu/Hu mice induced plasmablast formation and this activity was greater at later (>20 weeks) than earlier (<20 weeks) times post-transplant. However, Tfh-like cells from Hu/Hu mice demonstrated greater B cell helper function than those from Mu/Hu mice, particularly in the later time period. A similar trend was seen for Tph-like cells (*Figure 3D*), indicating that T-B cell interactions are more effective for T cells generated in a human thymus that is isogenic to the B cells than for those generated in a xenogeneic murine thymus, and consistent with previous literature (*Lang et al., 2013*; *Danner et al., 2011*; *Suzuki et al., 2012*).

Because B cells undergo class switching in germinal centers (GC), whose activity can be estimated by plasma CXCL13 levels, we analyzed serum concentrations of human CXCL13 in HIS mice. While human CXCL13 was detected in both groups and did not correlate with the number of splenic Tfh or Tph cells (data not shown), the levels tended to be higher in Mu/Hu compared to Hu/Hu mice (*Figure 3E*).

## Splenic B cell follicles and GC in HIS mice

We performed histological and immunostaining analyses of spleens to quantify B cell follicles in tissue sections from Mu/Hu (n=8) and Hu/Hu (n=11) mice in three time periods (<20 weeks, 20–30 weeks, and >30 weeks post-transplantation). Tissue sections were stained for CD3 (T cells), CD20 (B cells), and peanut agglutinin (PNA, marking GC-B cells) (*Figure 4A*). B cell follicles were manually quantified, and their size (in pixels per area) was determined using ImageJ software.

Hematoxylin and eosin (H&E) staining revealed a semiorganized splenic structure, with distinct B and T cell localization and well-defined follicles up to 30 weeks post-transplant in both groups. However, after 30 weeks, spleens showed fewer B cell follicles and appeared disorganized (*Figure 4B and C*). Three-color immunofluorescence staining on spleens from four Hu/Hu and three Mu/Hu mice at <20 weeks post-transplant revealed distinct B cell areas (CD20⁺), with PNA⁺ cells concentrated within B cell zones but also distributed in other splenic regions. Analysis of spleens from four Hu/Hu and three Mu/Hu mice at 20–30 weeks post-transplant revealed smaller B cell zones compared to earlier time points, and PNA⁺ cells were dispersed throughout the spleen rather than being enriched in B cell areas. Only two Mu/Hu mice showed clear B cell zones with localized PNA⁺ areas. Beyond 30 weeks post-transplant, spleens from two Hu/Hu and two Mu/Hu mice revealed no distinct B cell zones, and PNA⁺ cells were diffusely distributed.

Despite the increased number of Tfh-like cells in the spleens of Mu/Hu mice, there were no significant differences in the number or total area of splenic follicles between Mu/Hu and Hu/Hu mice (*Figure 4B*).

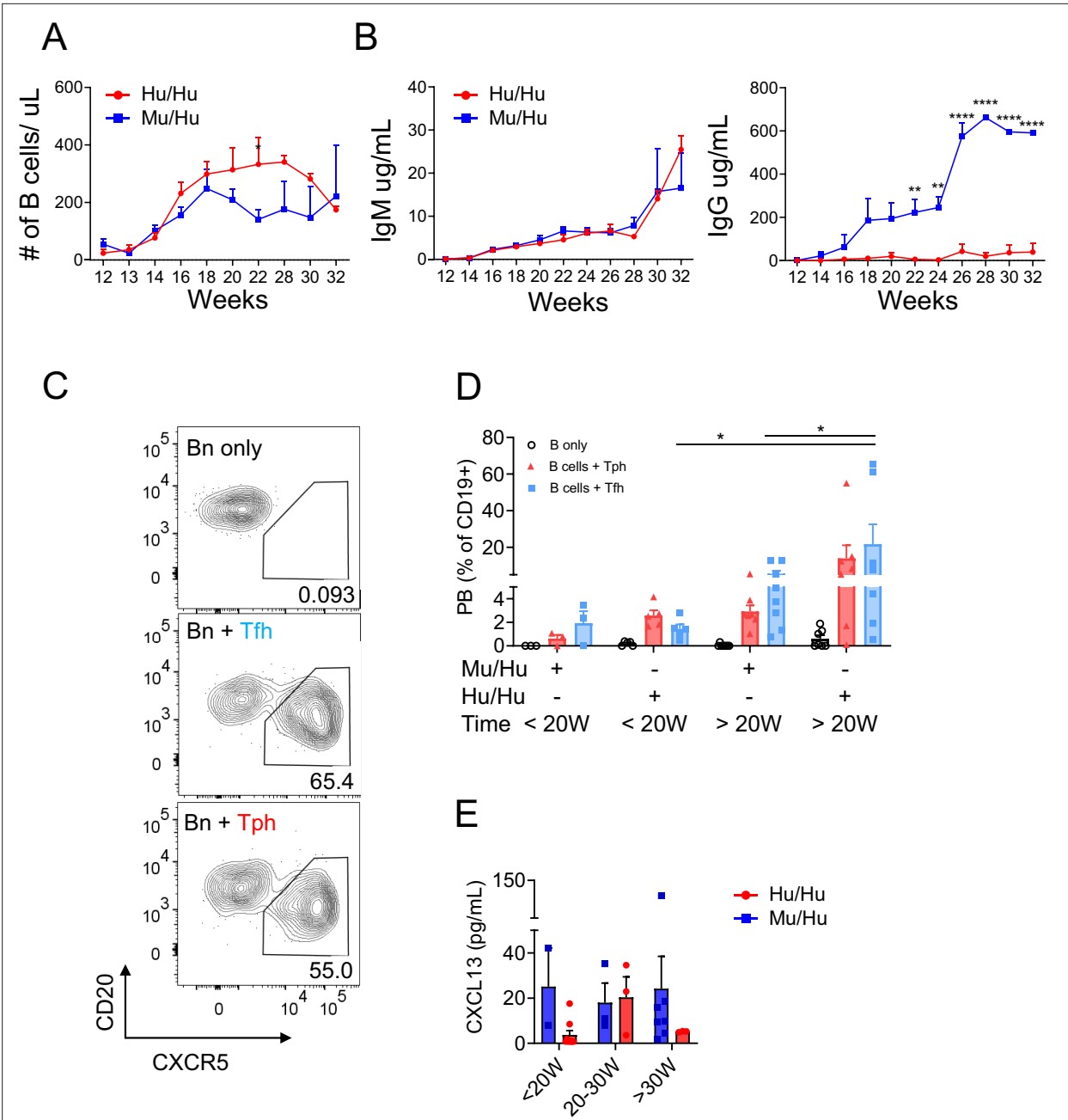

**Figure 3.** IgG and IgM antibodies, Tfh and Tph cell phenotypes, and B helper function of T cells from Mu/Hu vs Hu/Hu mice. (**A**) Absolute concentrations of CD19+ B cells in peripheral blood of human immune system (HIS) mice with mouse thymus (n=19) or human thymus (n=29) from week 12 to week 32 post-transplantation, analyzed every 2 weeks; and their (**B**) plasma IgM and IgG concentrations; (**C**) FACS-sorted Tfh (CD4+CD19-CD45RA-CXCR5+CD25-) and Tph (CD4+CD19-CD45RA-CXCR5-CD25-) cells from spleens of HIS mice with mouse vs human thymus were co-cultured with naïve B cells (CD4-CD19+CD38-IgD+CD27-) and staphylococcal enterotoxin B (SEB) as described in Materials and methods and plasmablast differentiation was assessed. (**D**) Percentage of naïve splenic B cells that differentiated into plasmablasts (CD4-CD19+CD20-CD38+) following co-culture with Tfh or Tph cells from HIS mice with mouse thymus (n=11) or human thymus (n=12) sacrificed at <20 weeks or >20 weeks post-transplantation. Splenocytes were obtained in the indicated time ranges. (**E**) Concentration of CXCL13 chemokine from plasma of HIS mice with mouse thymus (n=12) or human thymus (n=13). For (**A–B–G**), Bonferroni multiple comparison test was used. For (**C–D**) Friedman test was performed and corrected using Dunn's multiple comparisons test. For (**F**), Wilcoxon matched pairs signed rank test was used. Data are represented as mean ± SEM. *p<0.05, **p<0.01, and ****p<0.0001.

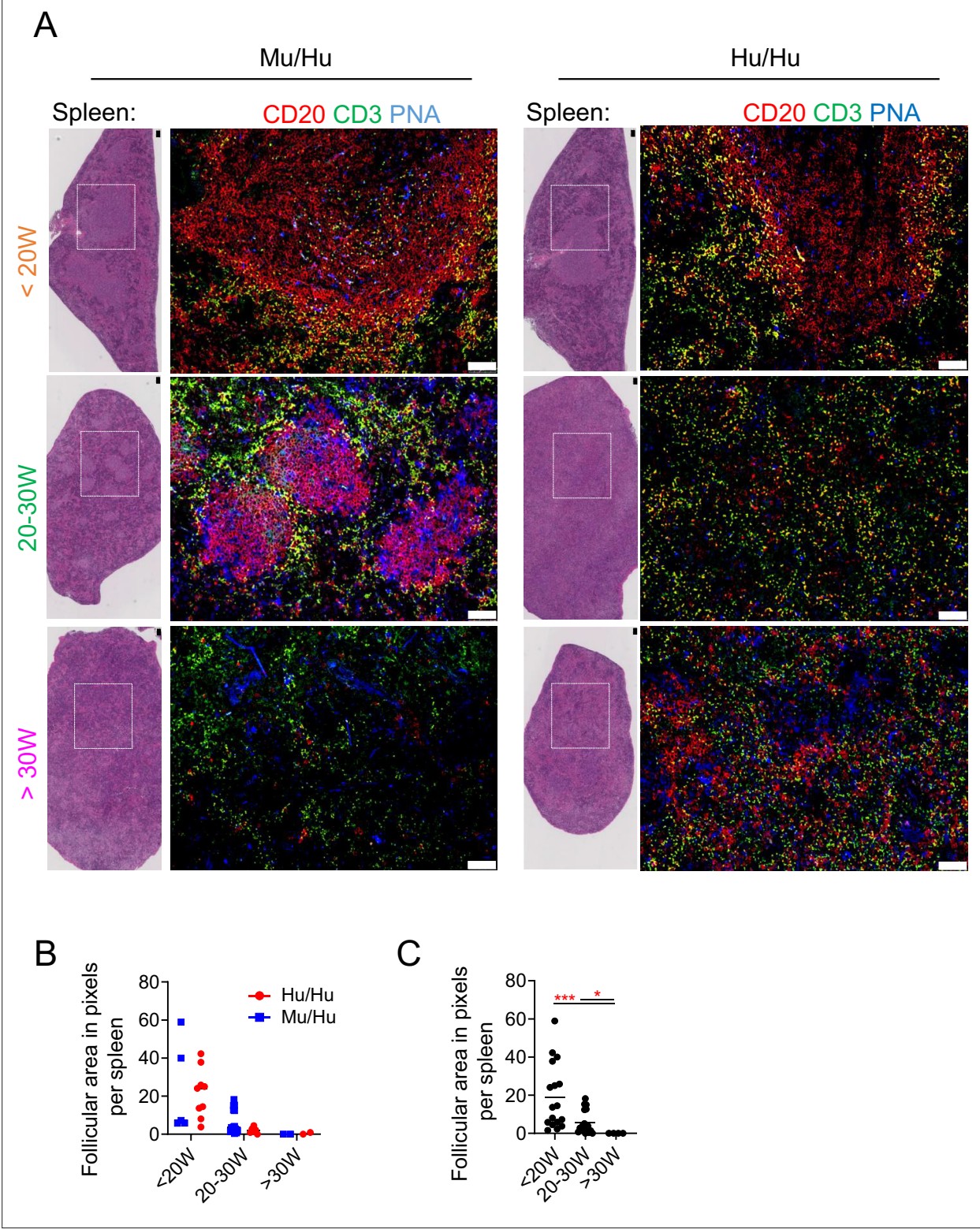

**Figure 4.** B cell follicles in human immune system (HIS) mice. (**A**) Hematoxylin and eosin (H&E) staining and immunofluorescence performed for CD20, CD3, and peanut agglutinin (PNA) in serial tissue sections of spleen from HIS mice with mouse vs human thymus at <20W, 20–30W, and >30W post-transplantation. Confocal images (10×) showing follicles (CD20⁺) and T cell zones (CD20⁻). (**B**) Quantification of follicular area in pixels between HIS mice with mouse vs human thymus. (**C**) Quantification of the numbers of follicles and follicular area in pixels over time, combining both groups of mice. Asterisks indicate statistical significance as calculated by Kruskal-Wallis. *p<0.05 and ***p<0.001. White squares in the H&E images indicate the area represented on the right side. White bar = 100 µm. An average of three different slides was examined per sample.

## Autoreactivity of IgM and IgG in HIS mice

To investigate the possible role of autoantibodies in driving autoimmune disease in HIS mice, we tested serum IgG and IgM for autoreactivity (*Figure 5*). *Figure 5B* confirms, in a separate experiment from that in *Figure 3*, a progressive increase in total serum IgM levels in both groups over time, while *Figure 5D* shows high levels of IgG antibodies in sera of Mu/Hu mice already by 20 weeks, when Hu/Hu mice still showed very low IgG levels. As shown in *Figure 5A* and *Table 1*, serum from Mu/Hu and Hu/Hu mice contained IgM antibodies that were reactive to multiple murine tissues. Several Mu/Hu mice also contained IgG antibodies with broad reactivity to multiple murine tissues. IgM in sera from HIS mice was reactive to LPS, insulin, and dsDNA, and these levels increased over time irrespective of the thymus type (*Figure 5C*). IgG against these antigens also increased over time in both groups (*Figure 5E*), even though total IgG levels tended to be higher in Mu/Hu than Hu/Hu mice (*Figure 5D*). To assess the possibility that IgM autoantibodies might be polyreactive, we compared the sum of IgM concentrations reacting to dsDNA, insulin, and LPS to total serum IgM concentrations and observed that the sum of these reactivities was >100% for three Mu/Hu mice (*Figure 5F*), consistent with polyreactivity of individual B cells in these mice.

## Depletion of B cells does not impact development of autoimmunity

To link the B cell types in HIS mice with autoantibody production, we performed phenotypic analysis of B cells. These studies revealed increased proportions of activated or age-associated B cells (IgD⁻CD27⁺CD11c⁺) and atypical B cells (IgD⁻CD27⁻CD11c⁺) in Mu/Hu mice compared to those from Hu/Hu mice (*Figure 5G*).

To assess the possible role of B cells in the development of autoimmunity in HIS mice, we treated Hu/Hu mice with rituximab (anti-CD20) once every 3 weeks from 20 to 38 weeks post-transplantation (*Figure 6A*). As shown in *Figure 6B*, this treatment successfully depleted B cells from the circulation and eliminated serum IgM. Immunofluorescent analysis of serum IgM from rituximab-treated mice showed an absence of autoreactive IgM (*Figure 6C*). However, B cell depletion with this method did not prevent or even delay disease development (*Figure 6D*), suggesting a minor, if any, role for antibodies in disease progression. However, since it remained possible that serum antibody was critical in presenting antigen to T cells and/or initiating inflammation that caused later disease, we also evaluated the impact of rituximab treatment early post-transplant. As shown in *Figure 6E*, starting rituximab treatment 1 week post-transplant and continuing through the post-transplant course also failed to delay or reduce the development of disease and, surprisingly, significantly accelerated its development. From these results, we concluded that antibodies were not required for disease development.

## Tfh- and Tph-like cells adoptively transfer disease in recipients containing human APCs without T cells: accelerated expansion and disease induction by T cells from Hu/Hu compared to Mu/Hu mice

To further clarify the role of Tfh- and Tph-like cells in the development of autoimmunity, we tested their ability to expand, differentiate, promote B cell differentiation, and induce disease in a T cell adoptive transfer model. We first generated a cohort of Mu/Hu and Hu/Hu mice from the same human FL CD34 cell donor. Twenty-two weeks later, after T cells had reconstituted the spleen, we euthanized both groups of HIS mice and FACS-sorted splenic CD4⁺CD45RA⁻CD45RO⁺PD-1⁺CXCR5⁺ Tfh-like and CD4⁺CD45RA⁻CD45RO⁺PD-1⁺CXCR5⁻ Tph-like cells. These T cells were adoptively transferred intravenously into thymectomized NSG mouse recipients that had received FL CD34⁺ cells 12 weeks earlier from the same human donor. Since these latter recipients lacked an endogenous (murine) thymus and had not received a thymus graft, they reconstituted only with B cells and myeloid APCs derived from the fetal HSC donor and did not generate T cells, as we have previously reported (*Khosravi-Maharlooei et al., 2021*; *Khosravi-Maharlooei et al., 2020*). Thus, there were five different recipient groups: (a) no T cell transfer (APC only, n=3), (b) mice that received Tph-enriched (n=5) or (c) Tfh-enriched cells (n=3) from Hu/Hu mice, and mice that received (d) Tph-enriched (n=5) and (e) Tfh-enriched cells (n=3) from Mu/Hu mice (*Figure 7A*). *Figure 7B* illustrates the gating strategy used to isolate Tfh and Tph cells from donor spleens.

As shown in *Figure 7C and D*, both Tfh-like and Tph-like cells originating in Hu/Hu mice expanded faster and to a greater extent than those originating in Mu/Hu mice, achieving higher circulating levels. Recipients of Tfh-like cells from Hu/Hu mice contained high numbers of circulating Tph-like and

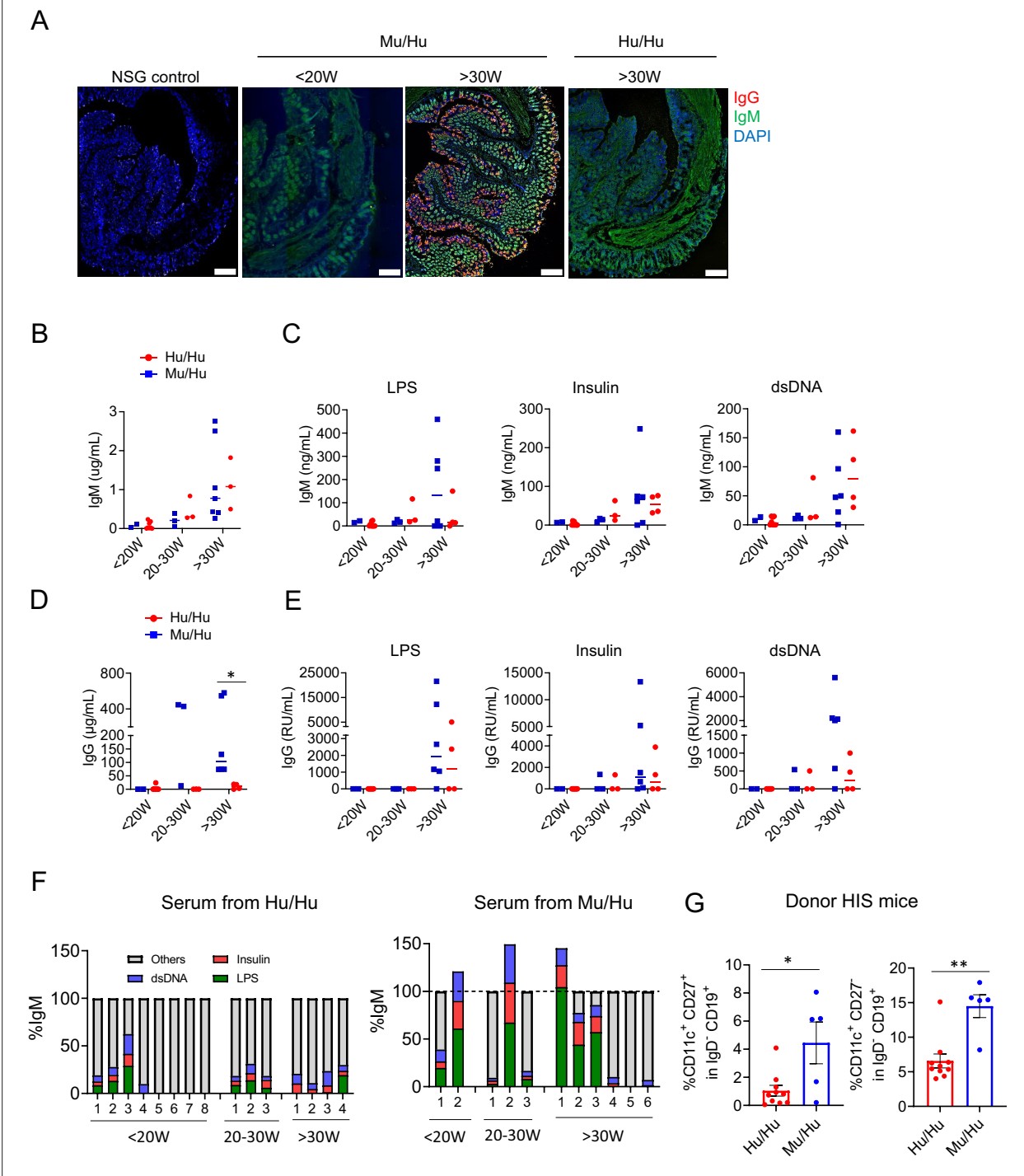

**Figure 5.** IgM and IgG antibodies from human immune system (HIS) mice are self-reactive. (**A**) NSG intestine stained with serum from HIS mice with mouse or human thymus or with naïve NSG mouse serum and secondary antibodies against human IgM and IgG. DAPI was used for nucleic acid staining. (**B–C**) Total concentration of serum IgM antibody and concentration of IgM antibody reactive to LPS, insulin, and dsDNA. (**D**) Total concentration of serum IgG antibody; (**E**) concentrations of IgG antibody reactive to LPS, insulin, and dsDNA. RU were defined as relative units compared to control supernatants from monoclonal polyreactive IgG-producing cell cultures. (**F**) Percentage of total serum IgM antibody that was reactive to LPS, insulin, and dsDNA from total IgM of HIS mice with mouse (n=11) vs human thymus (n=15) at <20W, 20–30W, and 30W post-transplantation. (**G**) Percentage of CD11c+ CD27+ and CD11c+ CD27+ in IgG- CD19+ B cells in the spleens of donor Hu/Hu and Mu/Hu mice. Asterisks indicate statistical significance as calculated by t-test *p<0.05. Means ± SEMs are shown.

**Table 1.** Mouse tissue-reactive human IgM and IgG in serum of Mu/Hu and Hu/Hu mice.

| | | | Pancreas | Liver | Spleen | Kidney | Thymus | SI | SG | Skin | AG | LI | Bone | Lung |
|---|---|---|---|---|---|---|---|---|---|---|---|---|---|---|
| Control mouse | NSG | IgM | – | – | – | – | – | – | – | – | – | – | – | – |
| | | IgG | – | – | – | – | – | – | – | – | – | – | – | – |
| Humanized, nothymus <20 W | Mouse #1 | IgM | + | + | + | + | + | + | + | + | + | + | + | + |
| | | IgG | – | – | – | – | – | – | – | – | – | – | – | – |
| | Mouse #2 | IgM | + | + | + | + | + | + | + | + | + | + | + | + |
| | | IgG | – | – | – | – | – | – | – | – | – | – | – | – |
| Mouse thymus <20 W | Mouse #3 | IgM | + | + | + | + | + | + | + | + | + | + | + | + |
| | | IgG | – | – | – | – | – | – | – | – | – | – | – | – |
| | Mouse#4 | IgM | + | + | + | + | + | + | + | + | + | + | + | + |
| | | IgG | – | – | – | – | – | – | – | – | – | – | – | – |
| | Mouse #5 | IgM | + | + | + | + | + | + | + | + | + | + | + | + |
| | | IgG | + | + | + | + | + | + | + | + | + | + | + | + |
| Mouse thymus >30 W | Mouse #6 | IgM | + | + | + | + | + | + | + | + | + | + | + | + |
| | | IgG | + | – | – | + | + | + | – | – | + | + | + | + |
| | Mouse #7 | IgM | + | + | + | + | + | + | + | + | + | + | + | + |
| | | IgG | – | – | – | – | – | – | – | – | – | – | – | – |
| | Mouse #8 | IgM | + | + | + | + | + | + | + | + | + | + | + | + |
| | | IgG | – | – | – | – | – | – | – | – | – | – | – | – |
| Human thymus >30 W | Mouse#9 | IgM | + | + | + | + | + | + | + | + | + | + | + | + |
| | | IgG | – | – | – | – | – | – | – | – | – | + | – | – |
| | Mouse #10 | IgM | + | + | + | + | + | + | + | + | + | + | + | + |
| | | IgG | – | – | – | – | – | – | – | – | – | – | – | – |
| | Mouse #11 | IgM | + | + | + | + | + | + | + | + | + | + | + | + |
| | | IgG | – | – | – | – | – | – | – | – | – | – | – | – |

SI, small intestine; SG, salivary gland; AG, adrenal gland; LI, large intestine.

much lower numbers of Tfh-like cells (*Figure 7E and F*), suggesting that Tfh-like cells might have lost CXCR5 expression or that Tph-like contaminants expanded preferentially. Recipients of Tph-like cells from Hu/Hu mice also contained low numbers of Tfh-like cells and much higher fractions of Tph-like cells, whose numbers and percentages were similar to those seen in recipients of Hu/Hu Tfh-like cells (*Figure 7F*). Recipients of Mu/Hu Tfh-like or Tph-like cells contained much lower numbers of human T cells of both types (*Figure 7D–F*), but T cells in the group receiving Mu/Hu Tfh-like cells also demonstrated a predominant Tph-like phenotype (*Figure 7F*). Similar trends were seen in spleens at the time of sacrifice, though differences did not achieve statistical significance (*Figure 7—figure supplement 1*, top row).

These differences in Tfh-like and Tph-like cell expansion from Hu/Hu vs Mu/Hu donor mice led us to compare B cell differentiation in the adoptive recipients. Proportions of B cells among human CD45+ cells (*Figure 8A*) reflected the degree of dilution due to expansion of transferred T cells noted above (*Figure 7D*) and overall circulating B cell counts were similar between groups (*Figure 8A*). No significant difference in proportions of naïve (IgD), memory (CD27), and IgG class-switched B cells was observed between the four groups and the APC-only HIS mouse control group, although there was a trend toward increased IgG+ B cells in adoptive recipients of T cells at later time points (*Figure 8B*).

By 5 weeks following adoptive transfer, the blood of recipients of Tfh and Tph cells from Hu/Hu mice exhibited increased levels of CD11c+ B cells, an effector B cell population associated with

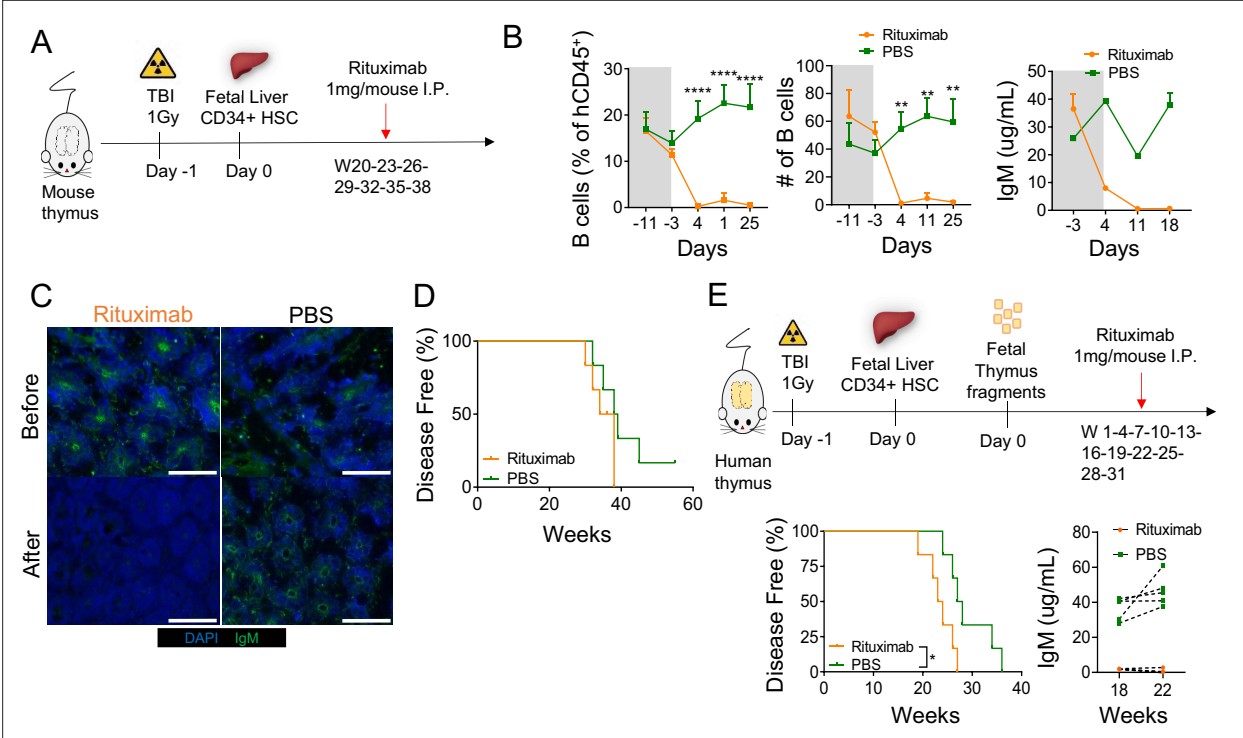

**Figure 6.** B cell depletion does not prevent disease development. (**A**) Human immune system (HIS) mice with mouse thymus were generated as described in Materials and methods and were injected intraperitoneally with 1 mg of rituximab (anti-CD20) or PBS every 3 weeks from W20 to W38. (**B**) Frequency and absolute number of CD19[+] B cells and serum IgM concentration before (gray) and after (white) treatment in HIS mice treated with rituximab or PBS (control). (**C**) NSG tissue stained with (primary) serum from HIS mice with mouse thymus and secondary antibodies against human IgM. DAPI was used for nucleic acid staining. (**D**) Kaplan-Meier curves for disease-free survival in relation to rituximab treatment in HIS mice with mouse thymus (n=6 per group). (**E**) Schema for early rituximab treatment initiation, Kaplan-Meier curves showing disease-free survival of each group, and serum IgM levels in rituximab-treated and PBS-treated control group (n=6 per group). All data are shown as means ± SEM. Asterisks indicate statistical significance as calculated by Bonferroni multiple comparison test. **p<0.01 and ****p<0.0001.

aging, infection, and human autoimmune diseases, compared to recipients of T cells from Mu/Hu mice (*Figure 8C*). This difference largely reflected an increase in the percentage of atypical B cells (IgD[−]CD27[−]CD11c[+], DN2) in the recipients of Hu/Hu T cells. Analysis of splenic B cell populations showed similar trends (*Figure 7—figure supplement 1*). Both activated or age-associated B cells (IgD[−]CD27[+]CD11c[+]) and atypical B cells (IgD[−]CD27[−]CD11c[+]) were significantly increased in the spleens of recipients of Hu/Hu Tph and Tfh cells compared to recipients of Mu/Hu Tph and Tfh cells 9 weeks following adoptive transfer (*Figure 8C*). Splenic total and naïve B cell percentages were not statistically different between groups (*Figure 7—figure supplement 1*). However, spleens of recipients of Tfh-like and Tph-like cells from Hu/Hu mice contained higher percentages of memory B cells and of CD11c[+] B cells than those of Mu/Hu T cell recipients (*Figure 7—figure supplement 1*, bottom row).

We also analyzed the levels of IgM and IgG from these four groups of recipient HIS mice and APC-only controls. Recipients of Tfh-like or Tph-like cells from Hu/Hu mice had higher levels of IgM and IgG antibodies compared to recipients of Tfh-like or Tph-like cells from Mu/Hu mice and the APC-only controls contained IgM at low levels and undetectable levels of IgG (*Figure 8D*). Collectively, these results demonstrate a role for Tfh-like and Tph-like cells in inducing both IgM and IgG antibodies in HIS mice and show that T cells from Hu/Hu mice provide help for autologous B cell antibody responses more effectively than those from Mu/Hu mice.

Adoptive recipients of Tfh-like and Tph-like cells from Hu/Hu mice lost weight beginning at 7 and 9 weeks, respectively, whereas the recipients of Mu/Hu T cells did not show obvious weight loss (*Figure 8E*). Clinical disease was also more apparent in recipients of Hu/Hu T cells, with the most rapid onset in recipients of Hu/Hu Tfh-like cells. In contrast, most recipients of Mu/Hu T cells did not develop clinically apparent disease during the follow-up period (*Figure 8F*). Together, our data suggest that T cells developing in an autologous human thymus graft interact more effectively with

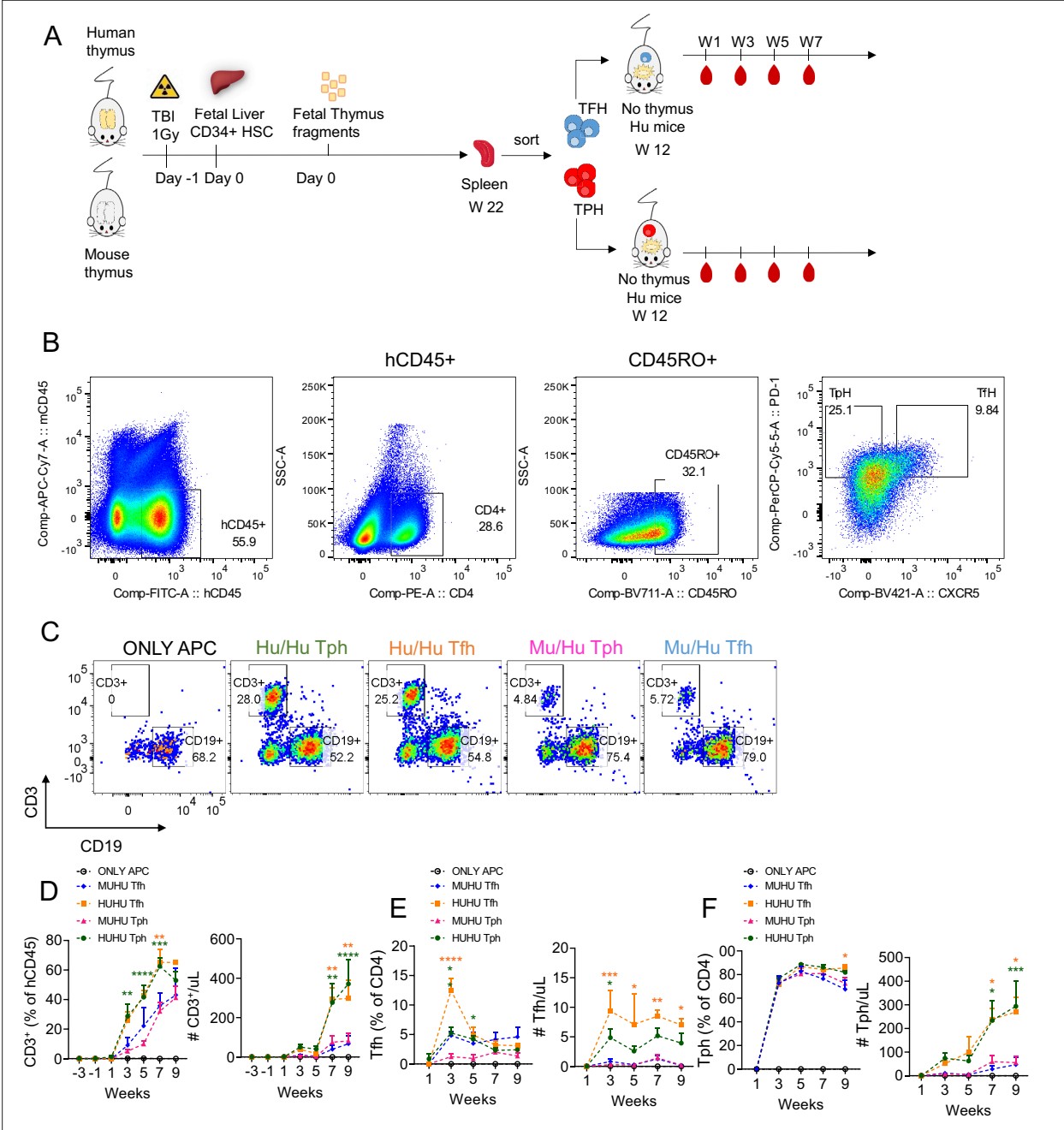

**Figure 7.** Expansion of donor T cells in thymecomized recipient human immune system (HIS) mice. (**A**) Schema for adoptive transfer experiment. Purified Tph or Tfh cells from reconstituted mice with mouse vs human thymus were adoptively transferred to thymectomized NSG mice that had received HSCs 12 weeks earlier from the same fetal liver CD34⁺ cell donor but did not receive a thymus graft. These mice therefore had B cells and other APCs but not T cells at the time of adoptive transfer. (**B**) The gating strategy used to isolate Tfh and Tph cells from donor spleens. (**C**) Representative plot of CD3⁺ T cells and CD19⁺ B cells in APC-only mice (n=*3*), adoptive recipients of Tph cells (n=*5*) or Tfh cells (n=*3*) from HIS mice with human thymus (Hu/Hu), or adoptive recipients of Tph cells (n=*5*) or Tfh cells (n=*3*) from HIS mice with mouse thymus (Mu/Hu). (**D**) Frequency and absolute number of CD3⁺ T cells. (**E–F**) Frequencies and absolute numbers of CXCR5⁺PD-1⁺ Tfh and CXCR5⁻PD-1⁻ Tph cells, respectively, in indicated groups. Asterisks indicate statistical significance as calculated by Bonferroni multiple comparison test. *p<0.05, **p<0.01, ***p<0.001, and ****p<0.0001. The green asterisks show significant differences between the Tph in Hu/Hu vs Mu/Hu mice, while the orange asterisks show significant differences between the Tfh in Hu/Hu vs Mu/Hu mice. Means ± SEMs are shown in panels D-F.

The online version of this article includes the following figure supplement(s) for figure 7:

**Figure supplement 1.** Frequencies of human T and B cell subsets in spleens of adoptive recipients of T cell subsets from Hu/Hu and Mu/Hu mice.

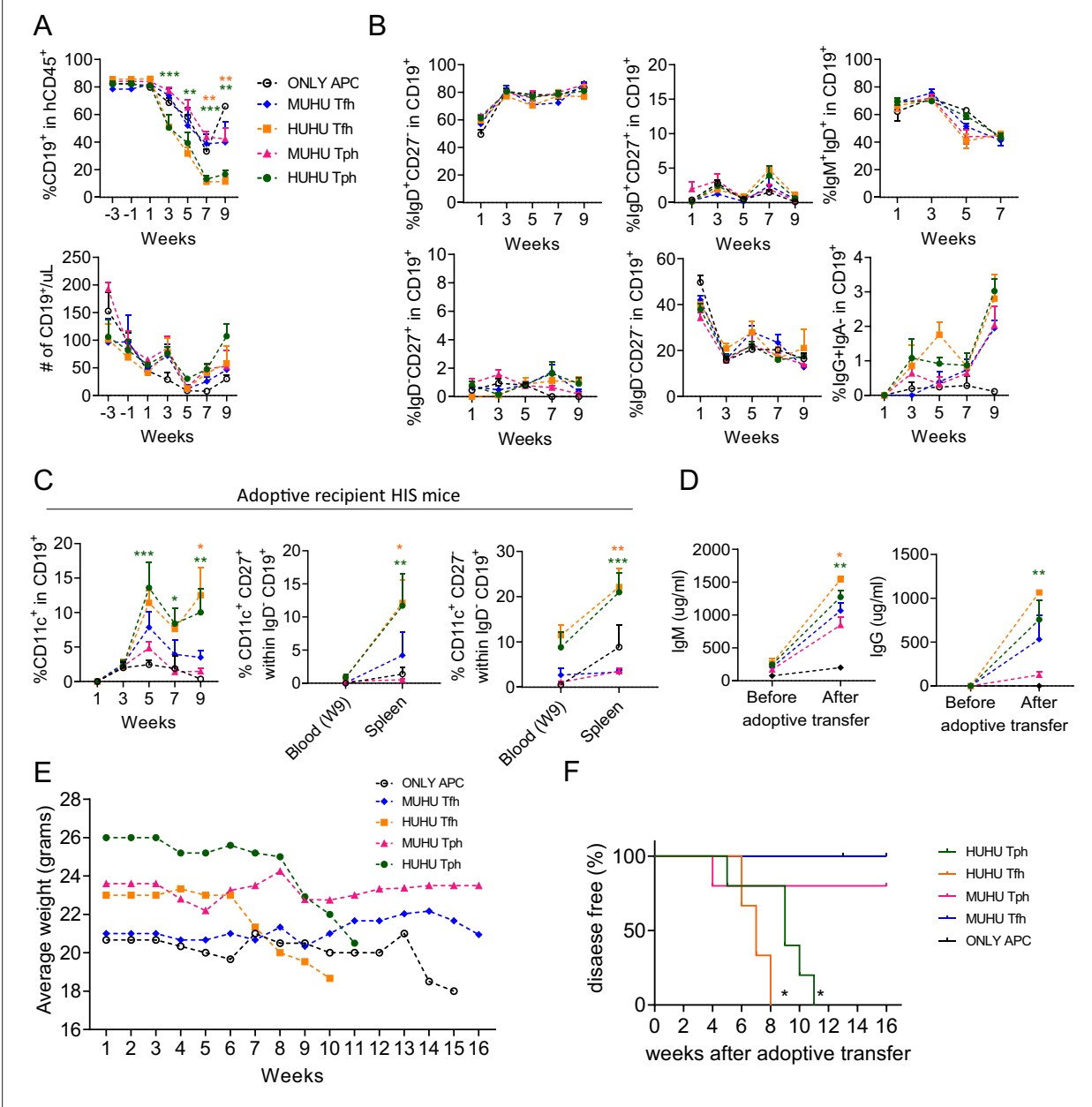

**Figure 8.** Effects of transferring Tfh and Tph from Mu/Hu and Hu/Hu mice to recipient human immune system (HIS) mice containing human APCs but not T cells. (**A**) Frequencies and absolute numbers of B cells, (**B**) percentages of IgD+CD27-, IgD+CD27+, IgD-CD27-, IgD-CD27-, IgM+IgD+, and IgG+ B cells among CD19+ B cells, (**C**) percentages of CD11c+ B cells among CD19+ B cells in the blood of APC-only mice (n=3), recipients of Tph cells (n=5) or Tfh cells (n=3) from HIS mice with human thymus (Hu/Hu), and recipients of Tph cells (n=5) or Tfh cells (n=3) from HIS mice with mouse thymus (Mu/Hu). Percentage of CD11c+ CD27 and CD11c+ CD27- in IgD- CD19+ B cells in the spleen and blood (W9 of adoptive transfer) of adoptive recipient mice. (**D**) IgM (left) and IgG (right) concentrations in sera of recipient mice before and after adoptive transfer. Means ± SEMs are shown in panels A-D. (**E**) Average weights over 16 weeks following adoptive transfer. (**F**) Kaplan-Meier curves for disease-free survival following injection of Hu/Hu Tph or Tfh or Mu/Hu Tph or Tfh cells. Animals were scored for autoimmune disease appearance every 2 weeks using a modification of published graft-vs-host disease (GVHD) scales (*Lai et al., 2012*; *Verlaat et al., 2022*) and considered to have disease if their score was >3 or if they showed >20% weight loss. Asterisks indicate statistical significance as calculated by Bonferroni multiple comparison test between Hu/Hu Tph and Mu/Hu Tph, or Hu/Hu Tfh and Mu/Hu Tfh cells. *p<0.05, **p<0.01, and ***p<0.001. Mantel-Cox test was used to analyze statistical significance in survival curve experiment.

peripheral B cells and other APCs, presumably reflecting thymic positive selection on the same HLA molecules as those in the peripheral APCs, resulting in T cell expansion, increased CD11c expression on B cells, and eventually disease development.

## Discussion

We recently demonstrated that human T cells play a requisite role in inducing a multiorgan auto-immune disease that develops in HIS mice that have a native murine or grafted human thymus and receive human CD34+ cells intravenously. The disease is characterized by organ infiltration by human T cells and macrophages and develops more rapidly in mice with a native murine thymus (Mu/Hu mice), where they fail to undergo normal negative selection, apparently due to the lack of a normal thymic structure, including a paucity of medullary epithelial cells (*Khosravi-Maharlooei et al., 2021*). In contrast, thymectomized HIS mice that receive human fetal thymus grafts (Hu/Hu mice) demonstrate negative selection for self-antigens (*Khosravi-Maharlooei et al., 2021*), have normal human cortico-medullary thymic structure and develop autoimmune disease significantly later (*Khosravi-Maharlooei et al., 2021*). The marked delay in disease development in Hu/Hu compared to Mu/Hu mice is mitigated by the presence of a murine thymus in non-thymectomized recipients of human thymus and CD34+ cells (*Khosravi-Maharlooei et al., 2021*), partially explaining the slower disease development in our model compared to studies in non-thymectomized 'BLT' mice (*Greenblatt et al., 2012*). Another reason for the delay in disease development in our model is the measures taken to deplete thymocytes preexisting in the thymus graft at the time of transplantation, as these also accelerate disease development (*Khosravi-Maharlooei et al., 2021*). Unlike GVHD induced by mature T cells transferred in human PBMCs (*Brehm et al., 2019*), the slowly evolving autoimmune disease (termed autoimmune because it is mediated by T cells developing de novo in the recipient rather than by cells carried in the graft) in Hu/Hu mice is independent of direct recognition of murine antigens, as it develops with similar velocity in NSG mice that express MHC and in those that completely lack (due to β2m and CIITA knockout) murine Class I and Class II MHC antigens (*Khosravi-Maharlooei et al., 2021*).

We have now demonstrated that autoimmunity in both Mu/Hu and Hu/Hu mice is associated with the appearance of Tph-like memory T cells and, to a greater extent in Mu/Hu than Hu/Hu mice, of Tfh-like cells in the circulation and spleen. Development of these T cell subsets is associated with differentiation of memory and class-switched B cells and characterized by the presence in serum of human IgM and IgG autoantibodies. The development of IgG autoantibodies was dependent on human T cells in HIS mice, consistent with previous literature (*Lang et al., 2013*; *Danner et al., 2011*; *Suzuki et al., 2012*). While mice lacking T cells also generated low levels of IgM autoantibodies, IgM levels were markedly increased by the presence of T cells.

Importantly, our data indicate that neither B cells nor antibodies are required for disease development. All HIS mice in our studies contained IgM autoantibodies, regardless of the presence or absence of T cells, though total IgM levels were increased in the groups containing T cells. In mice, the majority of natural IgM autoantibodies are produced by splenic B1 cells. Many of these are poly-reactive, demonstrate autoreactivity and may play a predominantly regulatory, protective role (*Lobo, 2016*). The high levels of IgM against specific antigens that were detected in sera of our HIS mice were consistent with polyreactivity, since the sum of their levels against three antigens was greater than total IgM levels in some cases. The lack of a functional C5 complement component, rendering membrane attack complex and C5a anaphylatoxin production impossible, may help to explain the lack of patho-genicity of autoantibodies observed in our studies and different results might be obtained in NSG recipients engineered to correct the C5 deficiency (*Verma et al., 2017*). However, C3 and C4 cleavage products of more proximal complement component activation might be expected to mediate signifi-cant pathology (*Goldberg and Ackerman, 2020*) and we are therefore surprised that B cells did not play a more readily apparent role in driving disease. In fact, the acceleration of disease in animals depleted of B cells from the beginning of the post-transplant period suggests that B cells may play a predominantly regulatory role in this model. Nevertheless, it is noteworthy that adoptive recipients of Tph-like and Tfh-like cells from Hu/Hu mice demonstrated an increase (compared to counterparts receiving T cells from Mu/Hu mice) in CD11c+ B cells, an effector B cell population otherwise known as age-associated B cells that is increased in association with human autoimmune diseases (*Mouat et al., 2022*; *Karnell et al., 2017*). These data directly demonstrate the capacity of Tfh- and Tph-like

cells from Hu/Hu, and to a lesser extent, from Mu/Hu mice, to induce the differentiation of this unique B cell subset. Our adoptive transfer studies also directly demonstrate the capacity of Tfh- and Tph-like cells generated in HIS mice to induce class-switched IgG-expressing and IgG-secreting B cells.

In vitro studies demonstrated effective help for differentiation of autologous B cells by T cells that differentiate in an autologous human thymus graft and less effective T-B interactions for T cells that differentiated in the xenogeneic murine thymus. These results are consistent with the interpretation that positive selection in an autologous thymus generates T cells that interact more effectively with autologous human APCs and B cells than T cells generated in a murine thymus lacking HLA molecules. Indeed, T cells have been shown to be essential for B cell maturation in HIS mice (*Lang et al., 2013*) and increased IgG levels are detected in association with greater T cell reconstitution (*Khosravi-Maharlooei et al., 2021*; *Lang et al., 2013*; *Brainard et al., 2009*). Furthermore, introduction of an HLA Class II molecule into the mouse has been reported to enhance human T-B cell interactions in HIS mice with a murine thymus (*Danner et al., 2011*; *Suzuki et al., 2012*).

In view of the improved T cell-B cell interactions in Hu/Hu compared to Mu/Hu mice, it is somewhat paradoxical that Mu/Hu mice demonstrate increased levels of class-switched IgG antibodies and increased activated/atypical B cells compared to Hu/Hu mice. However, we believe the failure to negatively select human autoreactive T cells in the murine thymus, as we have previously reported (*Khosravi-Maharlooei et al., 2021*), may provide an explanation for this paradox. If human T cells developing in a murine thymus are not centrally tolerized to murine antigens presented by human APCs, they may be expected to react with human APCs loaded with murine antigens, including human B cells, in the periphery in a manner that resembles an alloresponse. Alloresponses can activate B cells and induce polyclonal class-switched antibody responses that include autoantibodies, as shown in MHC heterozygous (F1) mice receiving MHC homozygous parental T cells, eliciting a GVH response that manifests as autoimmunity with autoantibodies, referred to as 'lupus-like' GVHD (*Soloviova et al., 2012*; *Tschetter et al., 2000*). Consistent with this possibility, responses of Mu/Hu splenic CD4 T cells to mouse antigen-loaded human DCs were stronger than those of Hu/Hu splenic T cells. Additionally, the increase in activated/age-associated (IgD⁻CD27⁺CD11c⁺) and atypical (IgD⁻CD27⁻CD11c⁺) B cells in Mu/Hu compared to Hu/Hu mice points to increased B cell activation in Mu/Hu mice.

The observed absence of a role for B cells in disease development thus requires alternative explanations for the more rapid disease development in Mu/Hu than Hu/Hu mice. Additional studies indicate that lymphopenia-induced proliferation (LIP) of T cells entering the periphery is increased in Mu/Hu compared to Hu/Hu mice, in which thymopoiesis and peripheral T cell reconstitution is more robust, and that LIP plays a significant role in the development of disease (M Khosravi-Maharlooei et al., manuscript in preparation). Consistently, studies in mice have strongly implicated LIP in autoimmune disease development (*King et al., 2004*; *Le Campion et al., 2009*; *McPherson et al., 2009*; *Zhang and Bevan, 2012*).

The adoptive transfer studies described here, in which Tfh- and Tph-like cells from Hu/Hu mice induced more rapid disease than those from Mu/Hu mice, demonstrated the relative disease-causing potential of these cells in the absence of variations in T cell reconstitution, as the cells were introduced to identical, completely T cell-deficient environments. Adoptive transfer of Hu/Hu T cells led to more rapid expansion than that of Mu/Hu T cells, suggesting that recognition of the same HLA antigens in the periphery as those mediating thymic positive selection optimizes LIP. This increased LIP may account for the more rapid disease development in the recipients of Hu/Hu than Mu/Hu Tfh- and Tph-like cells. Consistent with this interpretation, we have previously demonstrated that human peripheral APCs are absolutely required for survival, LIP, and effector differentiation of Hu/Hu T cells (*Onoe et al., 2010*). As reported for murine T cells (*Min et al., 2005*; *Surh and Sprent, 2008*), rapid LIP of human T cells is likely to be antigen-driven. Given that indirect presentation of murine antigens on human APCs is sufficient to drive disease (*Khosravi-Maharlooei et al., 2021*), we conclude that incomplete negative selection of indirectly xenoreactive mouse-specific T cells (possibly recognizing tissue-restricted antigens normally produced by murine thymic epithelial cells, which are absent in human thymus grafts) in a human thymus allows these T cells to recognize murine peptides presented on human APCs. This interpretation is consistent with our in vitro findings of tolerance to murine antigens presented directly on murine DCs, since murine APCs are present in the human thymic medulla of Hu/Hu mice (*Kalscheuer et al., 2012*) and of weak but measurable reactivity to autologous DCs capable of presenting murine antigens indirectly.

While lack of tolerance to murine antigens in the primary Mu/Hu mice reflects the lack of medullary structure and of central tolerance in the native mouse thymus (*Khosravi-Maharlooei et al., 2021*), the number of these T cells that expand in the periphery may be constrained by their positive selection on murine cortical epithelium and MHC, resulting in a more oligoclonal (compared to Hu/Hu mice) expansion of T cells recognizing murine peptides presented on human APCs expressing HLA. This situation is exacerbated by the markedly reduced diversity of the human TCR repertoire selected in the native murine thymus vs the human thymus graft (*Khosravi-Maharlooei et al., 2021*). The extensive expansion of a smaller population of peripheral T cells in Mu/Hu compared to Hu/Hu mice may have resulted in exhaustion of Tfh- and Tph-like cells in Mu/Hu mice prior to the time of transfer, delaying disease onset in adoptive recipients. Exhaustion has been reported for human T cells in HIS mice with GVHD (*Tary-Lehmann et al., 1994*) and chronic HIV infection (*Brainard et al., 2009*). The observed inability of direct antigen presentation on murine cells to drive LIP (*Onoe et al., 2010*) or disease development (*Khosravi-Maharlooei et al., 2021*), and hence dependence on antigen presentation on human APCs likely reflects failed coreceptor, costimulatory, adhesive, and/or cytokine interactions between human T cells and murine cells. While this in vivo result may seem inconsistent with the increased reactivity to murine antigens in vitro of T cells from Mu/Hu compared to those from Hu/Hu mice, human APCs were also present in these cultures, resulting in an inability to specifically assess direct xenoreactivity.

An additional possible contributor to accelerated autoimmune disease development in Mu/Hu compared to Hu/Hu mice may be inferior Treg function in the Mu/Hu group due to the incompatibility of the MHC of the murine thymic epithelium and the paucity of medullary epithelium responsible for positive selection of Tregs, resulting in a deficiency of Tregs that are capable of interacting with human APCs in the periphery or capable of recognizing murine antigens directly. This interpretation is consistent with the known role of Tregs in regulating Tfh activity (*Walker, 2022*).

Tfh and Tph are subsets of human effector T cells that are closely related and have been implicated in autoimmune diseases, yet have distinct phenotypes and functions, with Tfh providing help for naïve B cell differentiation and Tph acting predominantly on memory B cells (*Yoshitomi and Ueno, 2021*). Tfh are found largely in lymphoid tissues while Tph are generally found in inflamed tissues, though both seem to have a circulating component. Ratios of Bcl-6 to Blimp1 transcription factors are lower for Tph than Tfh and Tph have been suggested to drive extrafollicular differentiation of age-related B cells in autoimmune disease (*Yoshitomi and Ueno, 2021*; *Jenks et al., 2019*). While it is unknown whether there is plasticity/interchangeability between these subsets, our adoptive transfer studies are the first, to our knowledge, to suggest that there may be. In these studies, Tph-enriched T cells showed greater proliferative capacity overall than Tfh-enriched populations from Hu/Hu and Mu/Hu mice. Remarkably, the final number of Tph-like cells was quite similar and more than an order of magnitude higher than that of Tfh in recipients of either Tfh- or Tph-enriched populations, suggesting that Tfh may have the capacity to transition to the Tph phenotype when they expand in a lymphopenic environment. However, we cannot rule out the expansion of a contaminating Tph population in the sorted Tfh preparations, and fate-mapping studies will be needed to definitively address this question.

In summary, our data directly demonstrate the pathogenicity and capacity to induce antibody secretion of expanded human Tph- and Tfh-like cells and implicate both limitations in thymic selection and LIP as factors driving the development of autoimmune disease in HIS mice. Since Tfh, Tph, autoantibodies, and LIP have all been implicated in various forms of human autoimmune disease, the observations here provide a platform for the further dissection of human autoimmune disease mechanisms and therapies.

## Materials and methods
### Animals and tissues

NSG (NOD.Cg-*Prkdc*scid *Il2rg*tm1Wjl) mice (Strain #:005557, RRID:IMSR_JAX:005557) were purchased from Jackson Laboratory (Bar Harbor, ME, USA) and were housed and bred in specific pathogen-free helicobacter and Pasteurella pneumotropica-free conditions in the Animal Facility at Columbia University Medical Center. Human fetal thymus and FL tissues (gestational age 17–20 weeks) were obtained from Advanced Biosciences Resources. Fetal thymus fragments were cryopreserved in 10% dimethyl sulfoxide (Sigma-Aldrich) and 90% human AB serum (Gemini Bio Products). FL fragments were treated

**Table 2.** Graft-vs-host disease (GVHD) scoring system to assess disease severity based on five parameters: weight loss, posture, activity, fur condition, and skin condition.

| Parameter | Criteria | Score |
|---|---|---|
| | 10–25% weight loss compared to baseline | 1 |
| Weight loss | >25% weight loss compared to baseline | 2 |
| | Slight kyphosis | 0.5 |
| | Obvious kyphosis | 1 |
| | Kyphosis | 1.5 |
| Posture | Severe kyphosis | 2 |
| | Decreased activity | 0.5 |
| | Stationary >50% of the time | 1 |
| | Moves when stimulated | 1.5 |
| Activity | No movement | 2 |
| | Ventral ruffling | 0.5 |
| | Ventral line and slight back ruffling | 1 |
| | Ruffling >50% of the body | 1.5 |
| Fur condition | Ruffling entire body and denuded skin | 2 |
| | Flaking of ears, tail, or paws | 0.5 |
| | Erythema in tail or anus, ear shriveling | 1 |
| | Open lesions | 1.5 |
| Skin condition | Multiple open lesions | 2 |

for 20 min at 37°C with 100 µg/ml of Liberase (Roche) to obtain a cell suspension. Human CD34[+] cells were isolated from FL by density gradient centrifugation (Histopaque-1077, Sigma) followed by positive immunomagnetic selection using anti-human CD34 microbeads (Miltenyi Biotec) according to the manufacturer's instructions. Cells were then cryopreserved in liquid nitrogen. Studies were approved by the Animal Care and Use Committee at Columbia University under the IACUC protocol AC-AABM5551. All human samples were collected with approval of the Institutional Review Board of Columbia University Medical Center in accordance with the Declaration of Helsinki.

## Clinical definition of autoimmune disease

The definition of autoimmune-like disease after humanization with HSCs or following adoptive T cell transfer to T cell-deficient HIS mice is based on a GVHD scoring system described in our earlier publication. This system evaluates five criteria: percentage of weight loss, posture, activity level, fur condition, and skin condition, with each parameter assigned a grade from 0 to 2 (*Table 2*). For the disease-free survival curve (*Figures 6 and 8F*), a total score exceeding 3 or >20% weight loss was classified as indicative of disease.

## Human fetal tissue transplantation

To generate HIS mice with human thymus (Hu/Hu mice), 6- to 8-week-old NSG mice were thymecto-mized and allowed to recover for 2 weeks, then sublethally irradiated (1.0–1.2 Gy) using an RS-2000 X ray irradiator (Rad Source Technologies, Inc, Suwanee, GA, USA). Human fetal thymus fragments of about 1 mm³ were implanted beneath the kidney capsule and 2×10[5] human FL CD34[+] cells were injected IV. Recipient mice received intraperitoneal injections of 400 µg of anti-human CD2 mono-clonal antibody BTI-322 or Lo-CD2 26 (Bio X Cell, Inc) on days 0, 7, and 14 post-transplantation. HIS mice with intact mouse thymus (Mu/Hu mice) were generated simultaneously and received similar treatment without thymectomy or thymus transplantation.

## Flow cytometry

Human reconstitution, Tfh cells, Tph cells, and B cells were monitored in PBMC from 8 weeks after transplantation and in spleens of HIS mice until endpoint. Single-cell splenocyte suspensions and blood samples were treated with ACK lysis buffer (Life Technologies) to remove erythrocytes. After ACK erythrocyte lysis, remaining spleen cells were passed through a 70 µm filter prior to staining for FCM analysis. All cells were stained for detection of surface ICOS (ISA-3), CD45RA (HI100), CCR7 (G043H7), PD-1 (EH12.2H7), CXCR5 (RF8B2), CD8 (RPA-T8), CD25 (2A3), CD3 (SK7), CD4 (RPA-T4), CXCR3 (1C6), CD127 (A019D5), CCR2 (1D9), CCR5 (2D7), CD20 (L27), CD19 (HIB19), IgM (G20-127), IgG (G18-145), IgA (IS11-8E10), IgD (IA6-2), CD27 (O323), CD38 (HIT2), CD138 (MI15), CD11c (B-ly6), CD14 (M5E2), CD21 (Bu32), hCD45 (HI30), and mCD45 (30-F11). To detect IL-21, cells were stimulated for 3 hr with 50 ng/ml phorbol myristic acid (PMA; Sigma-Aldrich, St Louis, MO, USA), 1 µg/ml ionomycin (Sigma-Aldrich) and 3 µg/ml brefeldin A in RPMI supplemented with 10% FBS, 1% HEPES (Sigma-Aldrich), 1% penicillin-streptomycin (Life Technologies), and 0.05% gentamicin (Life Technologies) in 37°C, 5% $CO_2$ incubator conditions. For intracellular staining, single-cell suspensions were fixed-permeabilized with the FOXP3 Fixation Kit (Invitrogen) for 45 min at RT and then stained with mAbs for IL-21 (3A3-N2.1) for 30 min at RT in permeabilization buffer. After washing, cells were acquired on an Aurora Cytek (Fremont, CA, USA) or on a FACS FORTESSA (BD Biosciences), and analyzed with FlowJo v10 (Tree Star, USA) software.

## Proliferative responses of HIS mouse T cells to human and mouse antigens

Mu/Hu and Hu/Hu mice were sacrificed 20 weeks after transplantation and their splenocytes (not purified, so multiple murine and human cell populations were present) were CFSE-labeled and tested for reactivity to various antigen-presenting cells. To test direct reactivity to autologous human DCs, FL CD34$^+$ cells used to generate both Hu/Hu and Mu/Hu mice were differentiated into DCs. Briefly, HSCs were cultured in Aim-V medium containing 10% human serum and stem cell factor (SCF, 50 ng/ml) and granulocyte-macrophage colony-stimulating factor (GM-CSF, 50 ng/ml) for 3 days. After 3 days, additional GM-CSF (100 ng/ml) and SCF (50 ng/ml) and low dose TNF-α (1 ng/ml) were added to the culture. At day 7, cells were washed and cultured in medium containing GM-CSF (100 ng/ml), IL-4 (20 ng/ml), and TNF-α (2 ng/ml). One week later, differentiated DCs were harvested and either activated with prostaglandin E2 (PG-E2, 3000 ng/ml) and TNF-α (20 ng/ml) or incubated with apoptotic (irradiated at 30 Gy) NSG mouse DCs for 12 hr for indirect presentation of mouse antigens, followed by activation with PG-E2 and TNF-α. To generate NSG mouse DCs, mouse bone marrow cells were cultured in AIM-V media containing 10% FBS and murine IL-4 (5 µg/ml) and GM-CSF (3 µg/ml). After 1 week, mouse DCs were harvested and either activated with LPS (1 µg/ml) for 8 hr for direct presentation or irradiated (30 Gy) for co-culture with human DCs (for indirect presentation of mouse antigens on human DCs).

Splenocytes from Hu/Hu and Mu/Hu mice were stained with CFSE and cultured with various DCs (Auto human DCs, NSG DCs, Auto DCs loaded with mouse antigens, or no DCs) at a 10:1 ratio of splenocytes:DCs (100K CFSE-stained splenocytes and 10K DCs) in U-bottom 96-well plates. After 6 days, cells were stained with antibodies against human CD3, CD4, and CD8 and analyzed with a flow cytometer.

## T-B cell interaction assay

Single-cell splenocyte suspensions were stained with a panel to detect CD4 (RPA-T4), CXCR5 (RF8B2), CD45RA (HI100), CD19 (HIB19), IgD (IA6-2), CD27 (O323), and CD38 (HIT2) and prepared for sorting on a BD FACSAria Fusion II. Cells were first gated based on single cells and lymphocytes, and then Tfh cells (CD4$^+$CD45RA$^-$CXCR5$^+$CD25$^-$), Tph cells (CD4$^+$CD45RA$^-$CXCR5$^-$CD25$^-$), and naive B cells (CD19$^+$CD38$^-$CD27$^-$IgD$^+$) were FACS sorted. Cells were collected in RPMI 1640 (Life Technologies) supplemented with 10% FBS (Life Technologies). After collection, 20×10$^3$ Tfh cells or Tph cells and naïve B cells (CD19$^+$CD38$^-$CD27$^-$IgD$^+$) were incubated in the presence of staphylococcal enterotoxin B (0.1 µg/ml) in a 1:1 ratio (T cells:B cells). Cultures were performed in V-shaped 96-well plates in RPMI 1640 (Life Technologies) supplemented with 10% FBS, 1% HEPES (Sigma-Aldrich), 1% penicillin-streptomycin (Life Technologies), and 0.05% gentamicin (Life Technologies) in 37°C, 5% $CO_2$ incubator

conditions. After 7 days, cell pellets were harvested and stained to analyze plasmablast differentiation (CD19+CD20-CD38+).

## Plasma CXCL13 levels

Whole peripheral blood (in heparin) was centrifuged at 1000 rpm for 15 min to collect plasma. Plasma was further centrifuged at 13,000 rpm for 10 min to remove debris. CXCL13 was evaluated in EDTA plasma by ELISA assay (Human CXCL13/BLC/BCA-1 Quantikine ELISA Kit, R&D Systems) following the manufacturer's instructions.

## Plasma IgM and IgG levels

To quantify human antibodies in sera of HIS mice, diluted samples were added to plates (Corning Inc, Corning, NY, USA) coated with goat anti-human IgG Fcγ fragment (Jackson) or goat anti-human IgM (Southern Biotech) overnight at 4°C. Plates were then washed in PBS with 0.05% Tween 20, and blocked with 2% Bovine Serum Albumin (BSA, Fisher Scientific). Bound human Ig was detected using biotin-conjugated mouse anti-human IgG (BD Pharmingen) or biotin-conjugated mouse anti-human IgM (BD Pharmingen) secondary antibodies, followed by streptavidin-horseradish peroxidase (HRP) (Thermo Scientific). Colorimetric change by 3,3′,5,5′-tetramethylbenzidine substrate solution (Thermo Scientific) was stopped by 2 M sulfuric acid (Sigma), and optical densities determined by spectrophotometer absorbance at 450 nm. Human serum with known IgM and IgG concentrations (Bethyl) was used to establish standard curves.

## Assessment of reactivity to insulin, dsDNA, LPS molecules

To determine the reactivity of serum antibodies to insulin, dsDNA, and LPS, polystyrene plates (Corning) were coated overnight at 4°C with 10 µg/ml LPS, 10 µg/ml dsDNA, or 10 µg/ml recombinant human insulin (Sigma-Aldrich, St. Louis, MO, USA). After five washes in PBS with 0.05% Tween 20, serum was added and incubated for 2 hr at room temperature. Plates were then washed in PBS with 0.05% Tween 20, incubated for 1 hr with either HRP-conjugated goat anti-human IgM or IgG (Invitrogen, Camarillo, CA, USA), washed again, and developed using 3,3′,5,5′-tetramethylbenzidine (eBioscience, San Diego, CA, USA). Optical density was read at 450 nm absorbance. Supernatants from polyreactive IgM and IgG B cell cultures generated as described (*Porcheray et al., 2012*) and provided by Dr. Emmanuel Zorn (Columbia University) were used to establish standard curves.

## Histopathological analysis for HIS mouse models

For postmortem histopathology, spleen from HIS mice were excised, fixed in zinc-formalin (Sigma), and embedded in paraffin. Serial 5 µm sections were stained with H&E and images were acquired using Aperio AT2 (Leica Biosystems, Wetzlar, Germany).

## Immunofluorescence

Spleens from HIS mice were fixed with formalin and embedded in paraffin. FFPE tissues were cut using a Leica microtome at 5 µm. Cuts were done sequentially, placed on slides and left to dry for 24 hr prior to antibody staining. Tissues were deparaffinized and rehydrated by warming at 60°C for 1 hr followed by a serial bath of xylene and ethanol dilutions (100%, 95%, 80%, 70%, water). Antigen retrieval was performed using Borg RTU de-cloaking solution and a pressure cooker. The pressure cooker was set for 15 min at 11°C. Antigen retrieval was followed by permeabilization with confocal buffer for 1 hr. Titrated primary antibodies were added and left at 40°C overnight. Tissues were washed three times in 1× PBS and blocked with mouse and/or goat serum for 1 hr. Then, anti-human CD3 Alexa Fluor 488 (UCHT1, BioLegend, 1:50 dilution), anti-human CD20 Alexa 647 (Abcam, EPR1622Y), and peanut agglutinin (PNA/GC) antibodies were incubated for 1 hr at room temperature followed by three washes, then incubation with secondary antibodies for 2 hr. Tissues were washed again in 1× PBS followed by staining with nuclear marker DAPI and Fluoromount G mounting media.

For detection of mouse-reactive human IgM and IgG antibodies, sera from HIS mice were incubated on pancreas, liver, spleen, kidney, thymus, small intestine, large intestine, skin, bone, and lung sections taken from a naïve NSG mouse. After incubation for 2 hr at room temperature, serum was washed away and secondary anti-human IgM and IgG antibodies were added for detection for 2 hr

at room temperature. Negative controls were incubated with either no serum or naïve NSG mouse serum followed by secondary antibodies.

## B cell depletion experiments

A group of Mu/Hu mice were generated (*Figure 6A*) with irradiation (1 Gy) followed by injection with FL CD34+ HSCs. Starting at week 20 post-transplantation, mice were injected with rituximab (1 mg diluted in 1 ml of PBS, injected intraperitoneally) every 3 weeks until week 38. Blood was drawn at various time points before and after rituximab injection to measure human immune cell reconstitution in the blood and IgM levels in the serum. Mice were followed and evaluated for development of autoimmunity.

In another experiment (*Figure 6E*), Hu/Hu mice generated as described above received intraperitoneal injections of rituximab (1 mg/mouse/injection) starting from week 1 and every 3 weeks until week 31. Serum IgM was checked with ELISA and mice were followed and evaluated for the development of autoimmunity.

## Adoptive transfer experiment

Donor Mu/Hu and Hu/Hu mice were sacrificed at 22 weeks and CD4+CD45RA-CD45RO+PD-1+CXCR5+ Tfh and CD4+CD45RA-CD45RO+PD-1+CXCR5- Tph cells were FACS sorted from their pooled (for each donor type) splenocyte suspensions. $2.5 \times 10^5$ Tfh cells or Tph cells were adoptively transferred intravenously into recipient mice. The recipients were thymectomized NSG mice reconstituted 12 weeks earlier with the same HSCs as the Tfh/Tph donors, without a thymus transplant, which therefore contained autologous human APCs but no T cells. Adoptive recipient mice were monitored for disease and peripheral blood was collected from the tail vein once a week. After 9 weeks, adoptive recipient mice were euthanized. Spleens and blood were collected and stained for Tfh, Tph, and B cell analysis. Plasma was also collected for IgM and IgG evaluation.

## Statistical methods

Statistical analyses and comparisons were performed with GraphPad Prism 8.0 (GraphPad Software). All data are expressed as average ± standard error of mean. The nonparametric Mann-Whitney U test was used to compare two groups. One-way analysis of variance (ANOVA) with post hoc Tukey test was performed to compare three groups. Two-way ANOVA was used to resolve overall effects between transplant groups over time. When two-way ANOVA was significant, Bonferroni's multiple comparison test was run. Spearman's correlation coefficient was computed to study the strength of correlation between quantitative variables. Wilcoxon matched pairs signed rank test was used when analyzing paired groups. Mantel-Cox tests were used to measure significance of differences in Kaplan-Meier survival curves. $p < 0.05$ was considered to be statistically significant.

## Acknowledgements

We thank Dr. Remi Creusot for critical reading of the manuscript and Ms. Julissa Cabrera for assistance in its preparation. We also thank Dr. Emmanuel Zorn for providing control supernatants of cell lines expressing cloned polyreactive B cell receptors. This work was supported by NIH grants #P01 AI 045897 and U01 DK 123559.

## Additional information

### Funding

| Funder | Grant reference number | Author |
| --- | --- | --- |
| National Institutes of Health | P01 AI 045897 | Megan Sykes |
| National Institutes of Health | U01 DK 123559 | Megan Sykes |

| Funder | Grant reference number | Author |
|--------|------------------------|--------|

The funders had no role in study design, data collection and interpretation, or the decision to submit the work for publication.

## Author contributions

Mohsen Khosravi-Maharlooei, Conceptualization, Data curation, Formal analysis, Investigation, Visualization, Methodology, Writing – original draft, Writing – review and editing; Andrea Vecchione, Visualization, Writing – original draft, Writing – review and editing; Nichole Danzl, Data curation, Formal analysis, Investigation, Methodology, Writing – review and editing; Hao Wei Li, Robert Winchester, Writing – review and editing; Grace Nauman, Rachel Madley, Elizabeth Waffarn, Amanda Ruiz, Xiaolan Ding, Investigation; Georgia Fousteri, Conceptualization, Writing – review and editing; Megan Sykes, Conceptualization, Resources, Supervision, Funding acquisition, Writing – original draft, Project administration, Writing – review and editing

## Author ORCIDs

Mohsen Khosravi-Maharlooei (ID) http://orcid.org/0009-0001-2385-3714
Robert Winchester (ID) http://orcid.org/0000-0002-7543-8037
Amanda Ruiz (ID) http://orcid.org/0000-0002-6404-4905
Xiaolan Ding (ID) http://orcid.org/0009-0009-5220-1465
Megan Sykes (ID) https://orcid.org/0000-0002-4947-4376

## Ethics

These studies were performed in strict accordance with the recommendations in the Guide for the Care and Use of Laboratory Animals of the National Institutes of Health. The mice were purchased from Jackson Laboratory (Bar Harbor, ME) and were housed and bred in specific pathogen-free helicobacter and Pasteurella pneumotropica-free conditions in the Animal Facility at Columbia University Medical Center. The studies were approved by the Animal Care and Use Committee at Columbia University (IACUC protocol AC-AABM5551).

Reviewer #1 (Public review): https://doi.org/10.7554/eLife.99389.3.sa1
Reviewer #2 (Public review): https://doi.org/10.7554/eLife.99389.3.sa2
Author response https://doi.org/10.7554/eLife.99389.3.sa3

# Additional files

## Supplementary files

MDAR checklist

## Data availability

All data and the primary analysis files are publicly available at Zenodo.

The following datasets were generated:

| Author(s) | Year | Dataset title | Dataset URL | Database and Identifier |
|-----------|------|---------------|-------------|-------------------------|
| Maharlooei MK | 2025 | Figure 1- T cell proliferation fcs data | https://doi.org/10.5281/zenodo.15023230 | Zenodo, 10.5281/zenodo.15023230 |
| Maharlooei MK | 2025 | Figure 2 (manuscript title: Follicular helper- and peripheral helper-like T cells drive autoimmune disease in human immune system mice) | https://doi.org/10.5281/zenodo.15023369 | Zenodo, 10.5281/zenodo.15023369 |

*Continued on next page*

*Continued*

| Author(s) | Year | Dataset title | Dataset URL | Database and Identifier |
|---|---|---|---|---|
| Maharlooei MK | 2025 | Figure 4 (manuscript title: Follicular helper- and peripheral helper-like T cells drive autoimmune disease in human immune system mice) | https://doi.org/10.5281/zenodo.15023843 | Zenodo, 10.5281/zenodo.15023843 |
| Maharlooei MK | 2025 | Figure 6 (manuscript title: Follicular helper- and peripheral helper-like T cells drive autoimmune disease in human immune system mice) | https://doi.org/10.5281/zenodo.15024012 | Zenodo, 10.5281/zenodo.15024012 |
| Maharlooei MK | 2025 | Figure 7 and 8 (manuscript title: Follicular helper- and peripheral helper-like T cells drive autoimmune disease in human immune system mice) | https://doi.org/10.5281/zenodo.15024109 | Zenodo, 10.5281/zenodo.15024109 |
| Maharlooei MK | 2025 | Tables (manuscript title: Follicular helper- and peripheral helper-like T cells drive autoimmune disease in human immune system mice) | https://doi.org/10.5281/zenodo.15024036 | Zenodo, 10.5281/zenodo.15024036 |

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
