## [Editor Report · eLife Assessment]

This **important** study utilizes humanized mice, in which human immune cells are introduced into immune-deficient mice, to provide **convincing** evidence that two helper CD4 T-cell subsets, T-follicular helper (Tfh) and T-peripheral helper (Tph) cells, are able to drive both autoantibody production and induction of autoimmunity. The work will be of broad interest to medical scientists engaged in deciphering how human immune cells mediate immune responses and contribute to the development of autoimmune diseases.

---

## [Referee Report · Reviewer #1 (Public review)]

Summary:

As our understanding of the immune system increases it becomes clear that murine models of Immunity cannot always prove an accurate model system for human immunity. However, mechanistic studies in humans are necessarily limited. To bridge this gap many groups have worked on developing humanised mouse models in which human immune cells are introduced into mice allowing their fine manipulation. However, since human immune cells will attack murine tissues, it has proven complex to establish a human-like immune system in mice. To help address this Vecchione et al, have previously developed several models using human cell transfer into mice with or without human thymic fragments that allow negative selection of autoreactive cells. In this report they focus on the examination of the function of the B-helper CD4 T-cell subsets T-follicular helper (Tfh) and T-peripheral helper (Tph) cells. They demonstrate that these cells are able to drive both autoantibody production and can also induce B-cell independent autoimmunity.

Strengths:

A strength of this paper is that currently there is no well-established model for Tfh or Tph in HIS mice and that currently there is no clear murine Tph equivalent making new models for the study of this cell type of value. Equally, since many HIS mice struggle to maintain effective follicular structures Tfh models in HIS mice are not well established giving additional value to this model.

Weaknesses:

A weakness of the paper is that the models seem to lack a clear ability to generate germinal centres in which Tfh may exert some of their key functions. In some cases, the definition of Tph-like does not seem to differentiate well between Tph and highly activated CD4 T-cells in general, partly since the literature around these cells has not fully resolved this point.

---

## [Referee Report · Reviewer #2 (Public review)]

Summary:

Humanized mice, developed by transplanting human cells into immunodeficient NSG mice to recapitulate the human immune system, are utilized in basic life science research and preclinical trials of pharmaceuticals in fields such as oncology, immunology, and regenerative medicine. However, there are limitations to use humanized mice for mechanistic analysis as models of autoimmune diseases due to the unnatural T cell selection, antigen presentation/recognition process, and immune system disruption due to xenogeneic GVHD onset.

In the present study, Vecchione et al. detailed the mechanisms of autoimmune disease-like pathologies observed in a humanized mouse (Human immune system; HIS mouse) model, demonstrating the importance of CD4+ Tfh and Tph cells for the disease onset. They clarified the conditions under which these T cells become reactive using techniques involving the human thymus engraftment and mouse thymectomy, showing their ability to trigger B cell responses, although this was not a major factor in the mouse pathology. These valuable findings provide an essential basis for interpreting past and future autoimmune disease research conducted using HIS mice.

Strengths:

(1) Mice transplanted with human thymus and HSCs were repeatedly executed with sufficient reproducibility, with each experiment sometimes taking over 30 weeks and requiring desperate efforts. While the interpretation of the results is still debateble, these description is valuable knowledge for this field of research.

(2) Mechanistic analysis of T-B interaction in humanized mice, which has not been extensively addressed before, suggests part of the activation mechanism of autoreactive B cells. Additionally, the differences in pathogenicity due to T cell selection by either the mouse or human thymus are emphasized, which encompasses the essential mechanisms of immune tolerance and activation in both central and peripheral systems.

Weaknesses:

(1) In this manuscript, such as Fig. 2, the proportion of suppressive cells like regulatory T cells is not clarified, making it unclear to what extent the percentages of Tph or Tfh cells reflect immune activation. It would have been preferable to distinguish follicular regulatory T cells, at least. While Figure 3 shows Tregs are gated out using CD25- cells, it is unclear how the presence of Treg cells affects the overall cell population immunogenic functionally.

The authors added the data about FOXP3 expression among Tfh/Tph cells in the revised manuscript. This improved our data interpretation.

(2) The definition of "Disease" discussed after Fig. 6 should be explicitly described in the Methods section. It seems to follow Khosravi-Maharlooei et al. 2021. If the disease onset determination aligns with GVHD scoring, generally an indicator of T cell response, it is unsurprising that B cell contribution is negligible. The accelerated disease onset by B cell depletion likely results from lymphopenia-induced T cell activation. However, this result does not prove that these mice avoid organ-specific autoimmune diseases mediated by auto-antibodies and the current conclusion by the authors may overlook significant changes. For instance, would defining Disease Onset by the appearance of circulating autoantibodies alter the result of Disease-Free curve? Are there possibly histological findings at the endpoint of the experiment suggesting tissue damage by autoantibodies?

The authors appropriately modified the manuscript and provided sufficient information about the definition of diseases.

(3) Helper functions, such as differentiating B cells into CXCR5+, were demonstrated for both Hu/Hu and Mu/Hu-derived T cells. This function seemed higher in Hu/Hu than in Mu/Hu. From the results in Fig. 7-8, Hu/Hu Tph/Tfh cells have a stronger T cell identity and higher activation capacity in vivo on a per-cell basis than Mu/Hu's ones. However, Hu/Hu-T cells lacked an ability to induce class-switching in contrast to Mu/Hu's. The mechanisms causing these functional differences were not fully discussed. Discussions touching on possible changes in TCR repertoire diversity between Mu/Hu- and Hu/Hu- T cells would have been beneficial.

The authors correctly cited their previous findings about the TCR repertoire variation. This strengthened the discussion of this study.

---

## [Author Response]

The following is the authors’ response to the original reviews.

**Public Reviews:**

**Reviewer #1 (Public Review):**
Summary:As our understanding of the immune system increases it becomes clear that murine models of immunity cannot always prove an accurate model system for human immunity. However, mechanistic studies in humans are necessarily limited. To bridge this gap many groups have worked on developing humanised mouse models in which human immune cells are introduced into mice allowing their fine manipulation. However, since human immune cells will attack murine tissues, it has proven complex to establish a human-like immune system in mice. To help address this, Vecchione et al have previously developed several models using human cell transfer into mice with or without human thymic fragments that allow negative selection of autoreactive cells. In this report they focus on the examination of the function of the B-helper CD4 T-cell subsets T-follicular helper (Tfh) and T-peripheral helper (Tph) cells. They demonstrate that these cells are able to drive both autoantibody production and can also induce B-cell independent autoimmunity.Strengths:A strength of this paper is that currently there is no well-established model for Tfh or Tph in HIS mice and that currently there is no clear murine Tph equivalent making new models for the study of this cell type of value. Equally, since many HIS mice struggle to maintain effective follicular structures Tfh models in HIS mice are not well established giving additional value to this model.Weaknesses:A weakness of the paper is that the models seem to lack a clear ability to generate germinal centres. For Tfh it is unclear how we can interpret their function without the structure where they have the greatest influence. In some cases, the definition of Tph does not seem to differentiate well between Tph and highly activated CD4 T-cells in general.

The limited ability of HIS mice to generate well-defined lymphoid tissue structures is well noted. While the emergence of T cells in HIS mice increases the size of lymphoid tissues, the structure remains suboptimal and vaccination responses are limited. We believe this is mainly due to the common gamma chain knockout, which results in a lack of murine lymphoid tissue inducer (LTi) cells, which require IL-7 signaling to interact with murine mesenchymal cells for normal lymphoid tissue development. Ongoing efforts by our group and others aim to address this challenge by providing the necessary signals. Despite this challenge, these mice do develop Tfh cells, allowing us to study this cell subset.

We agree with the reviewer that the distinction between Tph and highly activated CD4 T cells is incomplete.

However, we have provided several distinctions in our manuscript that support the presence of Tph in HIS mice: (1) Tph cells exhibit very high levels of PD-1 expression, whereas other activated CD4 cells have varying levels of PD-1 expression. (2) Tph cells express IL-21. (3) Tph cells promote B cell differentiation and antibody production.

**Reviewer #2 (Public Review):**
Summary:Humanized mice, developed by transplanting human cells into immunodeficient NSG mice to recapitulate the human immune system, are utilized in basic life science research and preclinical trials of pharmaceuticals in fields such as oncology, immunology, and regenerative medicine. However, there are limitations to using humanized mice for mechanistic analysis as models of autoimmune diseases due to the unnatural T cell selection, antigen presentation/recognition process, and immune system disruption due to xenogeneic GVHD onset.In the present study, Vecchione et al. detailed the mechanisms of autoimmune disease-like pathologies observed in a humanized mouse (Human immune system; HIS mouse) model, demonstrating the importance of CD4+ Tfh and Tph cells for the disease onset. They clarified the conditions under which these T cells become reactive using techniques involving the human thymus engraftment and mouse thymectomy, showing their ability to trigger B cell responses, although this was not a major factor in the mouse pathology. These valuable findings provide an essential basis for interpreting past and future autoimmune disease research conducted using HIS mice.Strengths:(1) Mice transplanted with human thymus and HSCs were repeatedly executed with sufficient reproducibility, with each experiment sometimes taking over 30 weeks and requiring desperate efforts. While the interpretation of the results is still debatable, these description is valuable knowledge for this field of research.(2) Mechanistic analysis of T-B interaction in humanized mice, which has not been extensively addressed before, suggests part of the activation mechanism of autoreactive B cells. Additionally, the differences in pathogenicity due to T cell selection by either the mouse or human thymus are emphasized, which encompasses the essential mechanisms of immune tolerance and activation in both central and peripheral systems.Weaknesses:(1) In this manuscript, for example in Figure 2, the proportion of suppressive cells like regulatory T cells is not clarified, making it unclear to what extent the percentages of Tph or Tfh cells reflect immune activation. It would have been preferable to distinguish follicular regulatory T cells, at least. While Figure 3 shows Tregs are gated out using CD25- cells, it is unclear how the presence of Treg cells affects the overall cell population immunogenic functionally.

We analyzed the % FOXP3+ cells and the % of ICOS+ cells within the Tfh and Tph cells in the spleen of Hu/Hu and Mu/Hu mice at 20 weeks post-transplantation. Importantly, we see no difference in FOXP3 expression between Tfh of Mu/Hu and Hu/Hu mice. The results have been added to panels J and K of Figure 2.

(2) The definition of "Disease" discussed after Figure 6 should be explicitly described in the Methods section. It seems to follow Khosravi-Maharlooei et al. 2021. If the disease onset determination aligns with GVHD scoring, generally an indicator of T cell response, it is unsurprising that B cell contribution is negligible. The accelerated disease onset by B cell depletion likely results from lymphopenia-induced T cell activation. However, this result does not prove that these mice avoid organ-specific autoimmune diseases mediated by auto-antibodies and the current conclusion by the authors may overlook significant changes. For instance, would defining Disease Onset by the appearance of circulating autoantibodies alter the result of Disease-Free curve? Are there possibly histological findings at the endpoint of the experiment suggesting tissue damage by autoantibodies?

We have added a definition of disease to the Methods section as requested. Regarding the possibility of antibody-mediated disease that may be missed by this definition, we acknowledge this point in the Discussion section. However, we also discuss the point that the deficient complement pathway in NSG mice is likely to have protected the HIS mice from autoantibody-mediated organ damage.

(3) Helper functions, such as differentiating B cells into CXCR5+, were demonstrated for both Hu/Hu and Mu/Huderived T cells. This function seemed higher in Hu/Hu than in Mu/Hu. From the results in Figure 7-8, Hu/Hu Tph/Tfh cells have a stronger T cell identity and higher activation capacity in vivo on a per-cell basis than Mu/Hu's ones. However, Hu/Hu-T cells lacked an ability to induce class-switching in contrast to Mu/Hu's. The mechanisms causing these functional differences were not fully discussed. Discussions touching on possible changes in TCR repertoire diversity between Mu/Hu- and Hu/Hu- T cells would have been beneficial.

Consistent with the reviewer’s suggestion, we have previously shown that the TCR repertoire in Mu/Hu mice is less diverse than that in Hu/Hu mice (Khosravi-Maharlooei M, et al., J Autoimmun., 2021). We believe that the narrowed TCR repertoire in the periphery of Mu/Hu mice, combined with the inadequate negative selection in the murine thymus reported in the paper cited above, results in selective peripheral expansion primarily of the few T cell clones that are cross-reactive with HLA/murine self peptide complexes presented by human APCs in the periphery. We have discussed the reasons why these cells, when transferred to secondary recipients containing the same APCs, might not be as active as the more diverse, HLA-selected T cell repertoire transferred from Hu/Hu mice. These possible reasons include exhaustion of the T cells in Mu/Hu mice, limited expression of the few targeted HLA-peptide complexes recognized by the narrow cross-reactive TCR repertoire of Mu/Hu T cells and the consequent relatively impaired T-B cell collaboration in these mice.

**Recommendations for the authors:**

**Reviewer #1 (Recommendations For The Authors):**
The authors note that they removed an outlier result from Figures 1 B & C. With only 4 mice it seems difficult to see exactly how they determined the result was an outlier. Presumably, it was quite different from the others but in such a small dataset removing data without a very clear statistical rationale seems likely to strongly influence the results.

We have revised Fig 1 to include the previously-deleted outlier mouse.

Figure 4. The authors describe the follicular area. Were they able to observe any GC-like structures in their data?From the examples, I can see that the PNA staining is sometimes diffuse but even if the authors felt they could not observe a distinct GC this should be stated and discussed in the text.

We now describe the three colors IF staining in more detail in accordance with this comment. We characterized 4 Hu/Hu and 3 Mu/Hu spleens earlier than 20 weeks post-transplant. In all of these mice, distinct B cell areas (CD20+) were obvious and PNA+ cells were more concentrated in the B cell zones. We stained 4 Hu/Hu and 3 Mu/Hu spleens from mice between 20-30 weeks post-transplant and found that B cell areas were smaller in all these spleens compared to those taken before 20-weeks post-transplant. PNA+ areas are also more diffusely distributed and are not enriched in the B cell areas. Only 2 Mu/Hu mice showed clear B cell zones with some enriched PNA+ areas in the B cell zones. Additionally, we stained 2 Hu/Hu and 2 Mu/Hu mice later than week 30 post-transplant. No distinct B cell areas were observed in any of the spleens of these mice and PNA+ cells were diffusely distributed.

In Figure 3E the authors sort CD25-CXCR5-CD45RA- CD4 T-cells as Tph. This does seem a very loose definition including essentially all non-naïve CD4 cells that are not Tregs or Tfh.

We agree with the reviewer that the distinction between Tph and highly activated CD4 T cells is incomplete.

However, we have provided several distinctions in our manuscript that support the presence of Tph in HIS mice: (1) Tph cells exhibit very high levels of PD-1, whereas other activated CD4 cells have varying levels of PD-1 expression. (2) Tph cells express IL-21. (3) Tph cells promote B cell differentiation and antibody production.

Tph is sometimes a hard cell type to separate from more general highly activated CD4 T-cells. The broad CXCR5PD1+ phenotype they have used is common in the literature and the authors have confirmed some enrichment of IL21 production by these cells. However, they should consider if there are ways of further confirming this by examination of other markers such as CCR2 and CCR5 or elimination of other effector identities such as Th1 and Th17 or PD1+ exhaustion phenotypes.

For this study, we chose to follow the commonly used definitions in the literature for Tph and Tfh cells. For this reason, we are careful to refer to “Tph-like” cells rather than Tph cells in this manuscript. Distinguishing Tph cells from other subsets of activated CD4 cells would require further studies such as single cell RNA seq, which we hope to be able to perform in the future with additional funding.

Figure 8. The authors perform some analysis of B-cell phenotypes looking at markers such as CD27, IgD in 8B, and CD11c in 8C. Why is CD11c considered in isolation? The level of expression of the other markers would change how this data would be interpreted e.g. IgD-CD27-CD11c+ = DN2/Atypical cells, IgD-CD27+CD11c+ = Activated or ageassociated, etc.

In response to this comment, we reanalyzed the splenic samples of the donor Mu/Hu and Hu/Hu mice and their adoptive recipients. Interestingly, in the T cell donors, the Mu/Hu B cells included greater proportions of activated/age-associated B cells (IgD-CD27+CD11c+) and atypical cells (IgD-CD27-CD11c+), compared to the Hu/Hu B cells. This is consistent with the increased disease, increased Tph/Tfh and increased IgG antibody findings in the primary Mu/Hu compared to Hu/Hu mice. These results have been added to Figure 5G. We performed a similar analysis in the blood (week 9) and spleen of adoptive recipient mice. These studies showed that activated/ageassociated B cells (IgD-CD27+CD11c+) and atypical cells (IgD-CD27-CD11c+) were significantly increased in the adoptive recipients of Hu/Hu Tph and Tfh cells compared to the adoptive recipients of Mu/Hu Tph and Tfh cells (Fig. 8C). These results are consistent with the disease, T cell expansion and antibody results in the adoptive recipients.

Data not shown occurs often in this manuscript. In some cases what is not shown is potentially important. The authors note in the text relating to Figure 7 that the "purity of the cell populations as assessed by FCM ranged from 56-60% (data not shown)". Those numbers are a little alarming. They are referring to the purity of the FCS sorted Tfh and Tph prior to transfer? Currently, some of the discussion of this paper is about the possibility of plasticity, with Tfh switching into a Tph phenotype. If the transferred cell populations are 56-60% pure I don't think it is possible to make any interpretation of plasticity.

We looked into this further and realized that the purity figure cited in the original manuscript was erroneous due to a misunderstanding on the part of the first author of a question from the senior author. Unfortunately, data on the purity of the FACS-sorted population was not saved. However, we have added panel B to Figure 7 to show the sorting strategy for Tfh and Tph cells. We agree that any discussion of plasticity between these cell types is speculative, as outgrowth of a minor population is possible even from well-purified sorted cells.

Minor points:Some graphs have issues with presentation; Figures 5D and 5E, split scale clips data points. 5F the color representing time would be better replaced with direct labels. 6C and 6C some distortion of text clipping other elements.

We changed 5D and 5E y axis scales to avoid cutting the data points. Also, we changed 5F labels. Distortion of text clipping and other elements in Fig 6E and 6A have been corrected.

The abbreviation LIP is used in the abstract without a clear definition until later in the text.

This abbreviation has been defined again in the text.

Generally, the discussion section is quite long.

We agree that the discussion is quite long, but the results are quite complex and require considerable discussion. We have attempted to be as concise as possible.

**Reviewer #2 (Recommendations For The Authors):**
SuggestionCan Supplementary Figures be merged into the mains for the convenience of readers? There is enough extra margin.

We prefer to keep the order of main and supplementary figures as they are.

There are some confusing results which I would recommend to make the additional explanation for readers. For example, about 10% of Hu/Hu CD3+ T cells reacted to Auto-DC in Figure 1B, but neither CD4+ nor CD8+ cells did in Figure 1C.

We have re-analyzed the data in Fig 1 and included the previously-deleted outlier mouse.

MinorFigure 3CThe figure legend does not explain the figure. Hu/Mu or Mu/Mu?

Both groups were combined in the figure, as the results were similar for both. The N per group is given in the figure legend. The same applies to figure 3D.

Figure 4B, 4CWhy were Hu/Hu and Mu/Hu data merged only in 4B? They should be discussed in the context of parallel comparison. Both y-axis labels are the same between B and C despite the legend saying differently.

We switched the order of Figure 4B and 4C, each of which serves a different purpose. Figure 4B aims to demonstrate the similarity between the two groups at each timepoint. Figure 4C combines the two groups in order to provide sufficient animal numbers to demonstrate the statistically significant changes over time.

Figure 5DThe axis label was missing and the uncertain bar emerged. The authors should replace it with the corrected one.

The axis and the bar in 5D have been corrected.

Figure 5FThe legend does not explain the figure. What are these numbers? Also, it is better if the authors add a detailed explanation to the manuscript about the reason why the sum of antibody titer represents the poly-reactivity of IgM in these mice.

The numbers in the previous version of the figure were eartag numbers, which we have now renumbered as animal 1,2,3, etc in each group. Please refer to the final paragraph of the "Autoreactivity of IgM and IgG in HIS Mice" section in the Results section for an explanation of IgM polyreactivity.

Fig. 7D-E etc.The definition of Asterisk is insufficient. Between what to what in the multiple comparisons?

The green asterisks show significant differences between the Tph in Hu/Hu vs Mu/Hu mice, while the orange asterisks show significant differences between the Tfh in Hu/Hu vs Mu/Hu mice. This has been added to the figure legend.

Figure 7 ~ Figure 8The legends on the figure are confusing due to the different order of figures. The scales are inappropriate in some figures. The readers cannot interpret the data from the unfairly compressed plots.

We made the plots bigger to make them readable and changed the order.

MethodsIn the description of B cell depletion Experiments, the authors should directly mention the figure number instead of "In the second Experiment ..."

We have corrected this in the Methods section.

There is no definition of how to define the "disease" onset.

This definition has been added to the Methods section.

Several undefined abbreviations: "LIP", "BLT" ...

We defined these in the text.